# Functional connectivity–based classification and subtyping of major depression for precision mental health: An ensemble graph neural network approach

Kaizhong Zheng[1], Hongbing Lu[2], Huaning Wang[3], Dewen Hu[4], Badong Chen[1]*, Baojuan Li [2]*

1 National Key Laboratory of Human-Machine Hybrid Augmented Intelligence, National Engineering Research Center for Visual Information and Applications, and Institute of Artificial Intelligence and Robotics, Xi'an Jiaotong University, Xi'an, Shaanxi, China, 2 School of Biomedical Engineering, Fourth Military Medical University, Xi'an, Shaanxi, China, 3 Xijing Hospital, Fourth Military Medical University, Xi'an, Shaanxi, China, 4 Department of Intelligence Science and Technology, College of Intelligence Science and Technology, National University of Defense Technology, Changsha, China

* chenbd@mail.xjtu.edu.cn (BC), libjuan@163.com (BL)

## Abstract

Major depressive disorder (MDD) remains clinically diagnosed based on subjective symptoms rather than objective neurobiological markers, which limits diagnostic accuracy and the ability to tailor treatment. We present an ensemble hybrid framework that integrates graph neural networks (GNN) with unsupervised clustering to classify and subtype MDD using resting-state functional connectivity (rs-fMRI) profiles. A GNN was trained to distinguish MDD from healthy controls using functional connectivity derived brain graphs, and the resulting subject level embeddings were clustered to uncover subtype structure. We evaluated the approach on two public multisite cohorts, REST-meta-MDD (China; N = 1,604; 17 sites) and SRPBS (Japan; N = 446; 4 sites), using leave-one-site-out cross-validation and cross-national transfer. The classifier achieved 0.73 leave-one-site-out accuracy on REST-meta-MDD and retained 0.78 sensitivity when transferred from the Chinese to the Japanese cohort, outperforming BrainIB and CI GNN under the same protocol. To mitigate site related confounds, we applied a standardized preprocessing pipeline and ComBat harmonization. Clustering consistently identified three MDD subtypes with distinct connectivity signatures involving the default mode network and cerebellum, the insula-cingulum temporal circuit, and frontostriatal circuitry. These findings provide a reproducible and biologically interpretable stratification of MDD. Prospective studies will be needed to link these subtypes to treatment response and other clinically meaningful outcomes.

**Data availability statement:** Deidentified and anonymized data were contributed from studies approved by local Institutional Review Boards. All study participants provided written informed consent at their local institution. Data of the REST-meta-MDD project are available at: http://rfmri.org/REST-meta-MDD. Data of the SRPBS are available at: https://bicr-resource.atr.jp/srpbsfc.

**Funding:** This work was supported by the National Natural Science Foundation of China (32541016 to BC, 62436005 to BC, U25A20540 to BC, 82330043 to HW, and 62473303 to LC (Liangjun Chen)). BC contributed to the conceptualization, writing (review & editing), and supervision of the project. HW was responsible for data curation and writing (review). LC had no role in study design, data collection and analysis, decision to publish, or preparation of the manuscript.

**Competing interests:** The authors have declared that no competing interests exist.

## Author summary

Depression affects hundreds of millions of people worldwide and is a leading cause of disability, yet diagnosis still relies on reported symptoms rather than brain-based measures. This symptom-based approach can lead to misdiagnosis, delayed treatment, and suboptimal outcomes. We developed a machine learning framework that uses brain connectivity patterns from MRI scans to detect depression and identify biologically distinct subtypes. Using data from more than 2,000 participants across China and Japan, our method not only accurately distinguished patients from healthy individuals but also revealed three reproducible subtypes linked to specific brain circuits. By enabling a more objective and biologically grounded classification, this work could improve treatment matching, reduce unnecessary interventions, and support more efficient use of healthcare resources. Ultimately, such approaches could help deliver personalized and biologically informed mental health care worldwide.

## 1. Introduction

As a debilitating mental disorder, major depressive disorder (MDD) is regarded as the world's most serious psychiatric disorder [1,2] and represents the second leading contributor to chronic disease [3,4]. However, the treatment efficacy is still unsatisfactory and responses have been divergent. Current diagnosis with the Diagnostic and Statistical Manual of Mental Disorders 5th edition (DSM - 5) is based on subjective symptoms and signs rather than the underlying biological mechanisms. As a result, patients can meet diagnostic criteria through different combinations of symptoms [5], contributing to pronounced within-diagnosis heterogeneity and overlap with other psychiatric conditions [6–9]. This heterogeneity has been widely recognized as a major obstacle to developing mechanism-informed stratification and improving treatment outcomes.

To address these limitations, precision psychiatry initiatives and the Research Domain Criteria (RDoC) framework have advocated for redefining mental disorders using neurobiological systems, particularly brain circuits, rather than relying solely on symptom clusters [10]. In this context, resting-state functional connectivity (FC) has emerged as a promising candidate biomarker, offering a system-level characterization of distributed neural circuits [11,12]. Prior studies have shown that FC patterns can support machine-learning-based discrimination between patients and controls and can reveal clinically meaningful subgroups. For example, neuroimaging-driven approaches have been used to identify circuit-based signatures in schizophrenia [13] and related disorders, and several influential studies in depression have reported FC-defined subtypes associated with symptom profiles and treatment response [5]. Collectively, these findings suggest that connectome-based models may help deconstruct symptom-based categories and move toward biologically grounded subtyping.

However, a critical barrier to translation is limited generalizability. Many neuroimaging classifiers are trained and evaluated within a single dataset, and performance often degrades when models are tested on unseen sites or independent cohorts [5,14–16]. This fragility is likely driven by both disorder heterogeneity and substantial site-related variability in data acquisition and preprocessing, including differences in scanner hardware, sequence parameters and nuisance signals [17]. Consequently, robust validation across multiple sites and cohorts has been proposed as a minimal requirement for clinically relevant neuroimaging biomarkers, yet remains challenging in practice [18]. Even when harmonization strategies are applied, residual site effects and cohort differences can still limit reproducibility of both classification performance and data-driven subtyping. Despite extensive research on Major Depressive Disorder (MDD) classification and the optimization of generalization performance [16,19–22], achieving satisfactory results remains challenging, limiting the clinical applicability of these methods.

These challenges motivate the development of frameworks that simultaneously prioritize interpretability and generalizability, while supporting both case–control discrimination and subtype discovery. Here we present an ensemble hybrid framework, EH-BrainGNN, which learns compact and interpretable connectome-derived signatures and leverages them for classification and subsequent data-driven subtyping. We evaluate generalization in a large multi-site dataset and further test cross-cohort transfer in an independent validation cohort. By emphasizing reproducibility across sites and populations, our work aims to advance neuroimaging-informed stratification of depression that complements symptom-based diagnosis and provides a more biologically grounded basis for future precision-treatment studies.

## 2. Method

### 2.1. Participants

We used two rs-fMRI datasets in the current study. The "principal dataset" included data from 1604 participants (848 MDDs and 794 HCs from 17 sites; Table 1) and the "independent validation dataset" included data from 446 participants (177 MDDs and 269 HCs from 4 sites; Table 1). The most data of "principal dataset" and "independent validation dataset" can be downloaded publicly from the DIRECT consortium (http://rfmri.org/REST-meta-MDD) and SRPBS dataset (https://bicr-resource.atr.jp/srpbsfc/). The demographic characteristics of participants in both datasets across different sites are

**Table 1. Demographic characteristics of participants in both datasets.**

| Dataset Characteristic | Principal dataset | | | Independent validation dataset | | |
|---|---|---|---|---|---|---|
| | MDD (n=828) | HC (n=776) | P-value | MDD (n=177) | HC (n=269) | P-value |
| Age, years, mean (std) | 34.33 (11.48) | 34.44 (13.04) | 0.860 | 44.09 (12.08) | 45.05 (14.43) | 0.464 |
| Education, years, mean (std) | 12.01 (3.39) | 13.61 (3.41) | <0.0001 | NA[a] | NA[a] | NA |
| Gender, male (%) | 301 (36.35) | 318 (40.98) | 0.863 | 89 (50.28) | 100 (37.17) | 0.006 |
| Motion, mean (std) | 0.07 (0.04) | 0.07 (0.04) | 0.766 | 0.17 (0.11) | 0.18 (0.13) | 0.301 |
| HAMD, mean (std) | 21.2±6.5[b] | NA[b] | NA | NA[c] | NA[c] | NA |
| BDI, mean (std) | NA[d] | NA[d] | NA | 28.94 (9.71) | 7.72 (6.52) | <0.0001 |

Two independent datasets were used for this study including: the principal dataset from the DIRECT consortium and the independent validation dataset from SRPBS dataset. MDD=Major depressive disorder; HC=healthy controls; HAMD=Hamilton depression scale; BDI=Beck Depression Inventory-II; NA=not available.

[a]Years of education not available for all independent validation dataset.

[b]HAMD scores not available for some subjects in the principal dataset.

[c]HAMD scores not available for all independent validation dataset.

[d]BDI scores not available for all principal dataset.

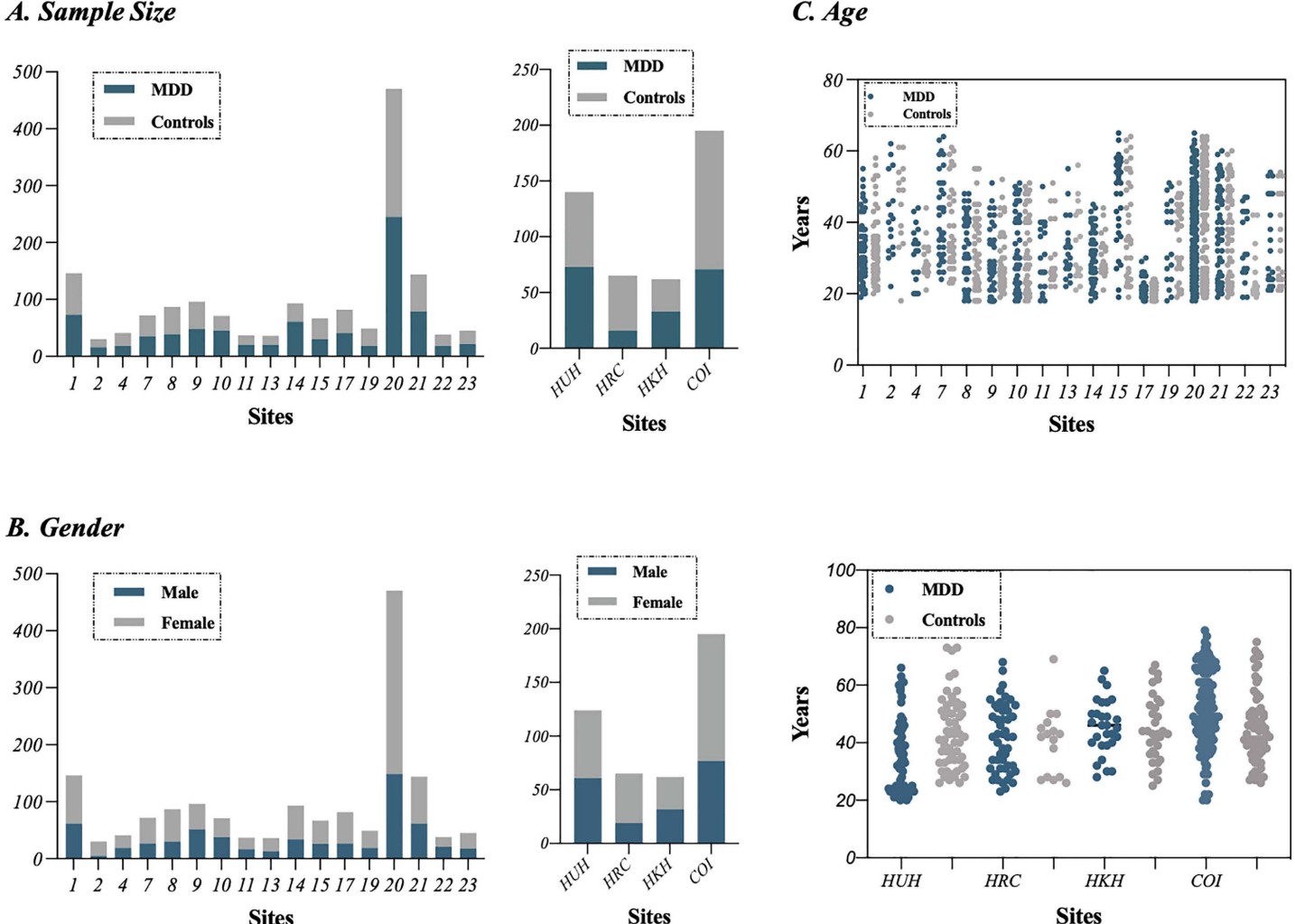

**Fig 1. Demographic characteristics of participants in both datasets across different sites.** The principal dataset contained 17 sites, while independent validation dataset contained 4 sites. **(A)** Total number of subjects for each contributing site in both datasets including the principal dataset (left side) and independent validation dataset (right side). **(B)** Number of female participants and male participants for each site in both datasets including the principal dataset (left side) and independent validation dataset (right side). **(C)** Age (in years) for all subjects per site in both datasets including the principal dataset (above side) and independent validation dataset (below side).

shown in Fig 1. Additional demographic details are provided in S1 Fig and S2 Fig, while the data acquisition parameters for the principal dataset are presented in Table C in S1 Text.

The principal dataset was provided from the DIRECT consortium which was launched in 2017 aiming to pooling neuroimaging data collected from multiple independent sites to boost the statistical power of data analysis and finally facilitate the clinical application of findings from neuroimaging studies. DIRECT consortium provided one of the largest MDD dataset (http://rfmri.org/REST-meta-MDD). Investigators from more than twenty-five research groups including our own had shared resting-state fMRI data. In the current study, 1604 participants (848 MDDs and 794 NCs) were selected according to exclusion criteria from a previous study [23] (S1 Fig).

The independent validation dataset was provided from the Japanese Strategic Research Program for the Promotion of Brain Science (SRPBS) dataset which included neuroimaging data of 2414 psychiatric patients and healthy controls

collected from eight Japanese universities and institutes. In this study, we selected 446 participants (117 MDDs and 269 NCs) from 4 sites including Hiroshima University Hospital (HUH), Hiroshima Rehabilitation Center (HRC), Hiroshima Kajikawa Hospital (HKH) and Hiroshima COI (COI).

## 2.2. Preprocessing

Standard preprocessing of the principal dataset and independent validation dataset were done using the Data Processing Assistant for Resting-State fMRI (DPARSF, http://rfmri.org/DPARSF/) and Graph Theoretical Network Analysis (GRETNA, https://www.nitrc.org/projects/gretna), which were based on Statistical Parametric Mapping (SPM, https://www.fil.ion.ucl.ac.uk/spm/software/spm12/). Standard preprocessing included discarding the initial 10 volumes, slice-timing correction, head motion correction, space normalization and temporal bandpass filtering (0.01-0.1 HZ). Statistical corrections removed effects of head motion, white matter and cerebrospinal fluid signals, as well as linear trends. Here, we adopted the Friston 24-parameter model to regress out head motion effects. After fMRI preprocessing, the brain was parcellated into 116 ROIs (regions of interest) according to the AAL (automated anatomical labelling) atlas [24]. The mean time courses of all the 116 ROIs were then extracted. We evaluated functional connectivity (Fisher's r-to-z transformed Pearson's correlation) between all ROI pairs. Therefore, a 116 × 116 symmetric matrix (functional connectivity network) $F$ was obtained for each subject.

## 2.3. Control of site differences and covariates

To mitigate site differences and confounding effects, we adopted ComBat harmonization method [25–28] to control for site differences and covariates in function connectivity. This method enabled us to preserve biological variability and subtract the variation introduced by site. We assumed that the fMRI data came from $m$ completely different multi-sites, including each $n_i$ participants for $i = 1, 2, \ldots, m$. For each functional connectivity, Combat model can be written as follows:

$$connectivity_{ij} = const + X_{ij}{}^T\beta + \gamma_i + \delta_i\varepsilon_{ij},$$

(1)

in which $connectivity_{ij}$ represents the FC value for the participant $j$ at site $i$ ($n \times 1$) and $const$ is the average FC value across all subjects from all sites, $X$ is a design matrix for the covariates of interest ($p \times n$, $p$ is the number of covariates including gender, age, education and head motion in the principal dataset, while $p$ includes gender, age and head motion in the replication dataset), $\beta$ is the vector of coefficients associated with $X$ ($p \times 1$), We further assume that the residual terms $\delta_i$ follow a normal distribution with mean zero and the terms $\gamma_i$ and $\delta_i$ represent the additive and multiplicative site effects of site $i$.

Next, the Combat estimates $\gamma_i^*$ and $\delta_i^*$ of the site effect parameters using uses Empirical Bayes. The final ComBat-harmonized functional connectivity is defined as:

$$connectivity^{Combat} = \frac{connectivity_{ij} - \widehat{const} - X_{ij}\hat{\beta} - \gamma_i^*}{\delta_i^*} + \widehat{const} + X_{ij}\hat{\beta},$$

(2)

in which $\widehat{const}$ represents the estimated average FC values. More detailed information has been previously described [25–27].

## 2.4. Problem definition

Given a set of weighted brain networks $\{G_1, G_2, \ldots, G_N\}$, the model outputs corresponding graph labels $\{y_1, y_2, \ldots, y_N\}$. Specifically, brain functional graph $G$ mainly contains the graph adjacency $A \in \{0, 1\}^{n \times n}$ characterizing the structure of graph and the node feature matrix $X \in \mathbb{R}^{n \times n}$ characterizing the feature of each node. Here, nodes are defined as regions

of interest (ROIs) based on brain parcellation. To create $A$, we binarized FC matrix by transforming only the top 20-percentile absolute correlation values into ones, and the rest were transformed into zeros. As for the node feature $X$, specifically $X_k$ for node $k$, it is defined as $X_k = [\rho_{k1}, \ldots, \rho_{kn}]T$, where $\rho_{kl}$ is the Pearson's correlation coefficient for node $k$ and node $l$. Fig 2A illustrates the pipeline from resting-state raw fMRI data to brain functional graphs. Generally, $N$ is defined as the number of subjects and $n$ is defined as the number of regions of interest (ROIs).

## 2.5. Overall framework of EH-BrainGNN

EH-BrainGNN follows a two-stage pipeline that decouples supervised representation learning from downstream subtyping (Fig 2 B and C and Fig 3). In the first stage, the model is trained to classify MDD versus healthy controls and, in doing so, learns subject-level graph representations. In the second stage, these learned graph-level features are used for unsupervised clustering to identify reproducible MDD subtypes. Importantly, the subtyping procedure operates on fixed embeddings and does not affect classifier training. To improve transparency, we provide a schematic of the complete deep

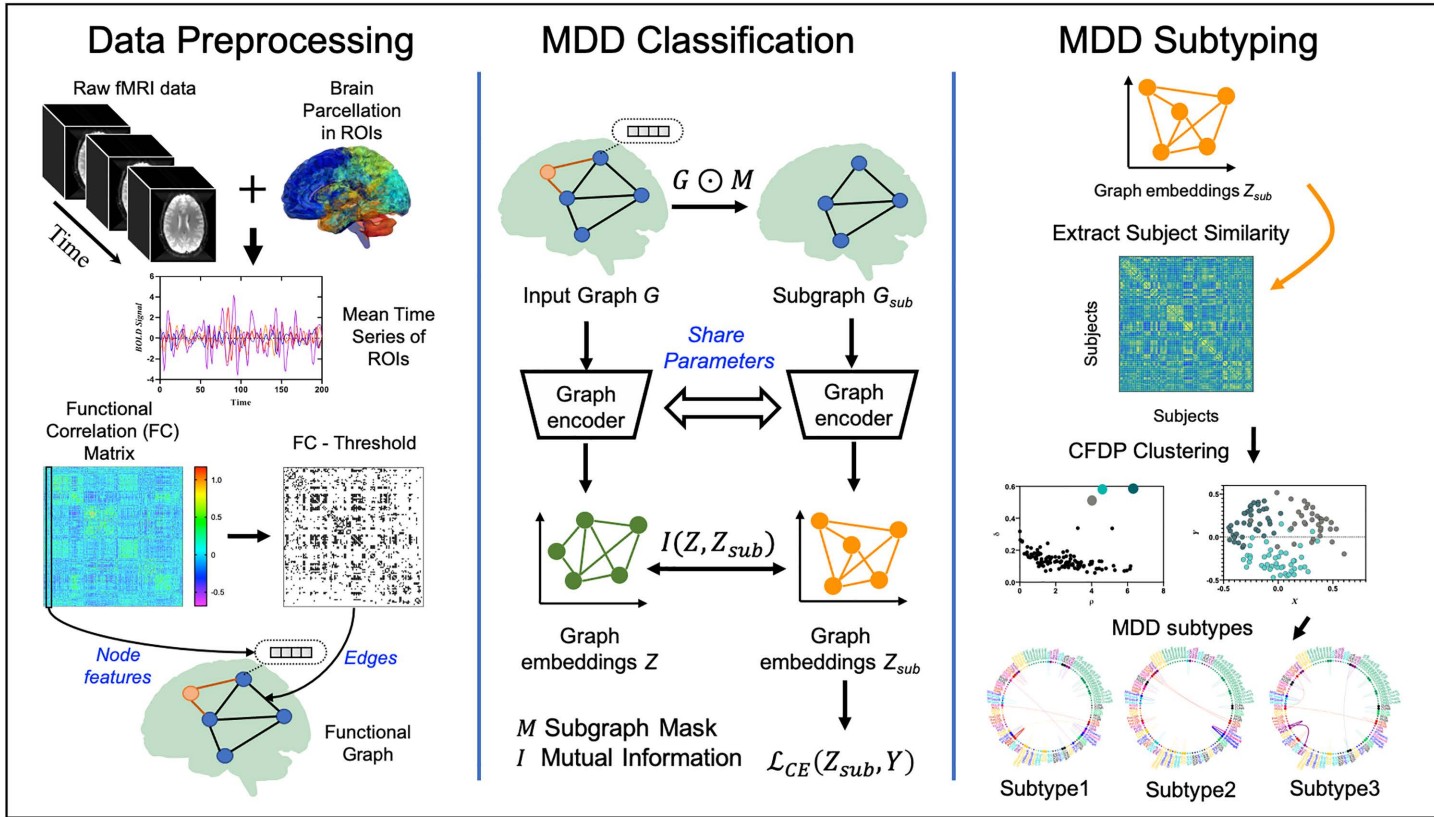

**Fig 2. Data analysis pipeline of EH-BrainGNN.** *Data Preprocessing:* Raw fMRI data were preprocessed and then mean time series of regions of interest (ROIs) were extracted from brain parcellation. Subsequently, the functional connectivity (FC) matrices were calculated using Pearson correlation between ROIs. From the FC we constructed the brain functional graph $G = (A, X)$, where $A$ was the graph adjacency matrix and $X$ is node feature matrix. In this study, $A$ was a binarized FC matrix where only the top 20-percentile absolute values of FC matrix were transformed into ones, while the rest were transformed into zeros. For node feature $X$, $X_k$ for node $k$ was an entire row of FC matrix. *MDD Classification:* We first learn subgraph mask $M$ to generate explanation subgraph $G_{sub} = G \odot M$, $\odot$ is element-wise multiplication. Then the graph encoder was employed to learn graph embeddings $Z$ and $Z_{sub}$ from $G$ and $G_{sub}$, respectively. Finally, we used $Z_{sub}$ to predict label $Y$. *MDD Subtyping:* The learned $Z_{sub}$ is used to extract subject similarity matrix by Pearson correlation between graph embeddings of every two participants. Then the subject similarity matrix was sent to the clustering by fast search and find of density peaks (CFDP) method to generate MDD subtypes.

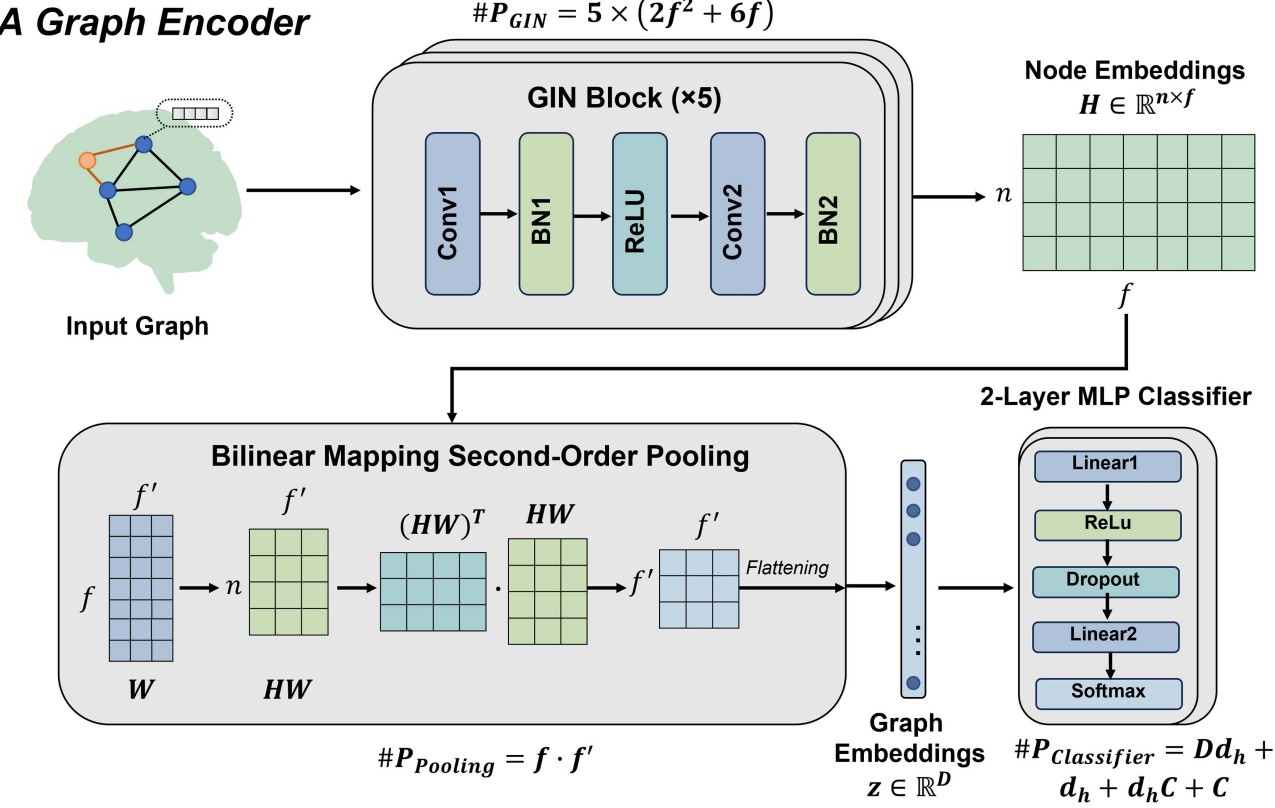

*A Graph Encoder*

$$\#P_{GIN} = 5 \times (2f^2 + 6f)$$

**GIN Block (×5)**: Conv1 → BN1 → ReLU → Conv2 → BN2

**Node Embeddings** $H \in \mathbb{R}^{n \times f}$

**Bilinear Mapping Second-Order Pooling**

$W$, $HW$, $(HW)^T$, $HW$, Flattening

$$\#P_{Pooling} = f \cdot f'$$

**Graph Embeddings** $z \in \mathbb{R}^D$

**2-Layer MLP Classifier**: Linear1 → ReLu → Dropout → Linear2 → Softmax

$$\#P_{Classifier} = Dd_h + d_h + d_h C + C$$

$n$: The number of nodes
$f$: The hidden dimensions of GIN
$f'$: The hidden dimensions of Bilinear Mapping Second-Order Pooling
$D$: The hidden dimension of graph embedding
$C$: The number of class
$d_h$: The hidden dimension of MLP layer

*B Subgraph Generator*

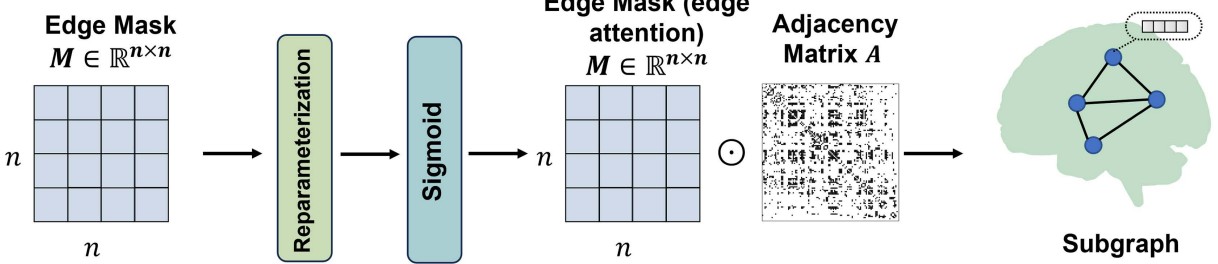

**Edge Mask** $M \in \mathbb{R}^{n \times n}$ → Reparameterization → Sigmoid → **Edge Mask (edge attention)** $M \in \mathbb{R}^{n \times n}$ $\odot$ **Adjacency Matrix** $A$ → **Subgraph**

**Fig 3. A schematic of the full network with parameter counts (A) and a clear statement of the role of the subgraph generator (B).**

network architecture and report the number of trainable parameters for each module and in total (Fig 3). Parameter counts are computed as the total number of trainable weights and biases in each learnable layer, whereas parameter free operations such as activation, dropout, pooling, and flattening are counted as zero.

**2.5.1. Stage 1 Supervised representation learning.** Given an input functional connectivity graph $G$, EH-BrainGNN learns discriminative representations under the supervised classification objective. This stage comprises three tightly coupled components.

(1) **Subgraph generator (explainer regularizer).** Subgraph generator learns a probabilistic edge mask and samples a discriminative subgraph $G_{sub}$ from the original graph $G$. The edge mask is optimized jointly with the classifier, encouraging the model to focus on task relevant connections and providing an interpretable set of edges that contribute most to the prediction.

(2) **Graph encoder.** The graph encoder processes either the full graph $G$ and the sampled subgraph $G_{sub}$ to produce node level embeddings and a compact graph level embedding via bilinear mapping second order pooling. The graph level embedding is then fed to a classifier head to predict the diagnostic label.

(3) **Mutual information estimator.** To prevent the subgraph generator from selecting spurious or overly sparse edges, we introduce a mutual information-based constraint between $G$ and $G_{sub}$. This term encourages $G_{sub}$ to preserve informative content from the original graph while remaining compact and discriminative. The subgraph generator and graph encoder are trained end to end with the combined classification loss and mutual information regularization.

**2.5.2. Stage 2 Downstream subtyping.** After training, we extract the graph level embeddings for all MDD subjects and perform clustering to derive subtypes. Because embeddings are produced by the supervised model, the resulting subtypes reflect heterogeneity in connectivity patterns that are relevant to MDD discrimination. This subtyping module is applied post hoc and does not alter the learned classifier.

## 2.6. Module A: Subgraph generator

The procedure of subgraph generator module is shown in Fig 3B. We firstly learned edge mask $M \in R^{n \times n}$ for each subject to extract the most informative or interpretable subgraph $G_{sub}$ from $G$. Specifically, the explanation subgraph $G_{sub} = (A', X)$ is induced by $M$, where $A' = A \odot M$, $\odot$ is element-wise multiplication.

To ensure the gradient, w.r.t., $M$ is computable, We learned $M$ using the reparameterization trick [29,30], where each element of $M$ represented the probability of each edge existing $e_{ij}$. Here, $e_{ij}$ could be calculated as:

$$\varepsilon \sim Uniform(0, 1), \quad e_{ij} = \sigma((\log \varepsilon - \log(1 - \varepsilon) + \omega_{ij})/\tau), \tag{3}$$

where $\sigma(\cdot)$ denotes the sigmoid function, $\omega_{ij} \in R$ is the learned parameter, and $\tau$ is a temperature for the concrete distribution.

Furthermore, to encourage the compactness of the explanation and the discreteness of $M$, we adopted $L_{sps}$ and $L_{ent}$ [31], respectively:

$$L_{sps} = \sum_{i,j} M_{i,j}, \quad L_{ent} = -(M \log(M) + (1 - M) \log(1 - M)). \tag{4}$$

## 2.7. Module B: Graph encoder

Graph encoder was mainly consisted of Graph Isomorphism Network (GIN) [32] encoder and the bilinear mapping second order pooling [33] (see Fig 3A). GINs use the $A$ and $X$ to learn the node embeddings $H$ through a neighborhood aggregation strategy. Specifically, we iteratively updated the representation of a node $h_v$ (*rows of H*) by aggregating and combining

the representations of neighboring nodes. In the first iteration, we initialized $H^{(0)} = X$, and $\mathcal{N}(v)$ was the set of neighboring nodes $v$. At the $k$-th iteration, $H^{(k-1)}$ are inputs and $H^{(k)}$ are outputs. Formally, the $k$-th layer of a GIN is

$$h_v^{(k)} = MLP^{(k)}\left((1 + \varepsilon^{(k)}) \cdot h_v^{(k-1)} + \sum_{u \in \mathcal{N}(v)} h_u^{(k-1)}\right)$$

(5)

where the multi-layer perceptron (MLP) is used to model and learn node representations, and $\varepsilon$ is a learnable parameter.

After obtaining the node embeddings, global graph pooling was applied to gain graph embeddings $h_G$ from node embeddings $H$. Recent work has proposed a novel global graph pooling methods based on second-order pooling for graph classification tasks (bilinear mapping second order pooling) [33]. Second-order pooling (SOPOOL) is described as:

$$\text{SOPOOL}(H) = \sum_{i=1}^{n} h_i h_i^T = H^T H \in R^{f \times f}$$

(6)

SOPOOL is able to effectively capture the correlations among features, and the encoded topology information. In contrast to conventional graph pooling approaches, bilinear mapping-based second-order pooling leverages information from all nodes, captures second-order statistical dependencies, and achieves parameter efficiency by substantially reducing the number of trainable variables.

Here, we used bilinear mapping second order pooling to obtain graph representations or graph embeddings $h_G$.

S2 Fig (b) shows an illustration of generating graph embeddings. First, we reduced the dimensionality of $H$ using a linear mapping:

$$\text{SOPOOL}_{bimap}(H) = \text{SOPOOL}(HW) = W^T H^T H W \in R^{f' \times f'}$$

(7)

where $W$ is a trainable matrix representing a linear mapping. Then we flatten the matrix and obtain a graph embedding:

$$h_G = FLATTEN(\text{SOPOOL}_{bimap}(H)) \in R^{f'^2}$$

(8)

Note that graph embedding $h_G$ is an $f'^2$-dimensional vector.

## 2.8. Module C: Mutual information estimator

We introduced the information bottleneck principle to obtain robust and compact subgraph structure. Specifically, we minimize the mutual information between the original graph $G$ and the explanation subgraph $G_{sub}$ and maximize the mutual information between subgraph $G_{sub}$ and label $Y$. Formally, the optimization framework is formulated as:

$$\min_M MI(G_{sub}, G) + \max_M MI(G_{sub}, Y)$$

(9)

where $MI$ denotes the mutual information and $M$ is an edge mask.

Based on a previous study [34], $\max_M MI(G_{sub}, Y)$ could be interpreted as a classic cross-entropy loss. Maximizing $MI(G_{sub}, Y)$ encourages $G_{sub}$ is most predictable to graph label $Y$. Mathematically, it could be defined as:

$$-MI(G_{sub}, Y) \leq \mathbb{E}_{Y, G_{sub}} - \log q_\theta(Y \mid G_{sub}) := L_{CE}(G_{sub}, Y),$$

(10)

in which $q_\theta(Y \mid G_{sub})$ is the variational approximation to the true mapping $p(Y \mid G_{sub})$ from $G_{sub}$ to $Y$.

In order to minimize $MI(G_{sub}, G)$, we first extract graph embeddings $Z_{sub}$ and $Z$ from the original graph $G$ and its subgraph $G_{sub}$ using the graph encoder. According to sufficient encoder assumption [35] that the information of $Z$ was lossless in the encoder precess, we further approximated $MI(G_{sub}, G)$ with $MI(Z_{sub}, Z)$, where $Z_{sub}$ and $Z$ represented graph embeddings of $G_{sub}$ and $G$ obtained from the graph encoder, respectively. Thus, Eq. (9) was changed as:

$$\max_M MI(G_{sub}, Y) \approx L_{clf} = -\frac{1}{N}\sum_{n=1}^{N}\left(y_n\log\left(\widehat{y}_n\right) + (1-y_n)\log\left(1-\widehat{y}_n\right)\right),$$
$$\min_M MI(G_{sub}, G) = L_{mask} = \min_M MI(Z_{sub}, Z),$$

(11)

where $N$ represents the number of participants, $y_n$ is label for the $n$-th subject, $\widehat{y}_n$ is the prediction of model for the $n$-th subject.

In addition, we used the recently proposed matrix-based Rényi's $\alpha - order$ mutual information [36,37] to directly estimate $MI(Z_{sub}, Z)$. It's mathematically well-defined and computationally efficient, and dosen't require an additional neural network unlike the mutual information neural estimator (MINE) [38]. Rényi's $\alpha - order$ entropy, defined via the normalized eigen spectrum of the Gram matrix corresponding to the data projected onto a reproducing kernel Hilbert space (RKHS), allows for direct estimation of $H_\alpha(Z)$, $H_\alpha(Z_{sub})$ and $H_\alpha(Z, Z_{sub})$ from the data itself. This approach eliminates the need for discrete probability density function (PDF) estimation or the use of auxiliary neural networks.

According to Shannon's chain rule, $MI(Z_{sub}, Z)$ can be decomposed as:

$$MI(Z_{sub}, Z) = H_\alpha(Z_{sub}) + H_\alpha(Z) - H_\alpha(Z_{sub}, Z),$$

(12)

where $H_\alpha(Z_{sub})$ indicates the entropy of $Z_{sub}$, $H_\alpha(Z)$ indicates the entropy of $Z$ and $H_\alpha(Z_{sub}, Z)$ represents the joint entropy between $Z_{sub}$ and $Z$. The specific calculation process is directly given by the definitions.

**Definition 1**. Let $\kappa : \chi \times \chi \mapsto \mathbb{R}$ be a real valued positive definite kernel that is also infinitely divisible. Given $\{x_i\}_{i=1}^{n} \in \chi$, each $x_i$ can be a real-valued scalar or vector, and the Gram matrix $K \in \mathbb{R}^{n \times n}$ calculated as $K_{ij} = \kappa(x_i, x_j)$, a matrix-based analogue to Rényi's $\alpha$-entropy can be given by the following functional:

$$H_\alpha(A) = \frac{1}{1-\alpha}\log_2(tr(A^\alpha)) = \frac{1}{1-\alpha}\log_2\left(\sum_{i=1}^{n}\lambda_i(A)^\alpha\right),$$

(13)

in which $\alpha \in (0, 1)\bigcup(1, \infty)$. $A$ is the normalized $K$, i.e., $A = K/tr(K)$. $\lambda_i(A)$ denotes the $i$-th eigenvalue of $A$.

**Definition 2**. Given a set of $n$ samples $\{x_i, y_i\}_{i=1}^{n}$, each sample includes two measurements $x \in \chi$ and $y \in \gamma$ obtained from the same realization. Given positive definite kernels $\kappa_1 : \chi \times \chi \mapsto \mathbb{R}$ and $\kappa_2 : \gamma \times \gamma \mapsto \mathbb{R}$, a matrix-based analogue to Rényi's $\alpha$-order joint-entropy can be defined as:

$$H_\alpha(A, B) = H_\alpha\left(\frac{A \circ B}{tr(A \circ B)}\right),$$

(14)

where $A_{ij} = \kappa_1(x_i, x_j)$, $B_{ij} = \kappa_2(y_i, y_j)$ and $A \circ B$ shows the Hadamard product between the matrices $A$ and $B$.

Given $\{Z^i, Z_{sub}^i\}_{i=1}^{N}$ in a mini-batch of $N$ samples, we first estimate two Gram metrics $K = \kappa(Z, Z) \in \mathbb{R}^{N \times N}$ and $K_{sub} = \kappa(Z_{sub}, Z_{sub}) \in \mathbb{R}^{N \times N}$ associated with $Z$ and $Z_{sub}$, respectively. According to Definitions 1 and 2, the entropy and joint entropy terms in Eq. (13) and (14), such as $H_\alpha(Z)$ and $H_\alpha(Z_{sub})$, can be simply computed over the eigen spectrum of $K$ and $K_{sub}$. Here, we use the radial basis function (RBF) kernel $\kappa$ to obtain $K$ and $K_{sub}$:

$$\kappa(z_i, z_j) = \exp(-\frac{\|z_i - z_j\|^2}{2\sigma^2}),$$

(15)

and for the kernel width $\sigma$, we estimate the mean of the $k$ (where $k = 10$) nearest distances for each sample.

Thus, the final loss function of EH-BrainGNN is described as:

$$L = L_{clf} + \lambda_1 L_{mask} + \lambda_2 L_{sps} + \lambda_3 L_{ent}, \tag{16}$$

where $\lambda_1$, $\lambda_2$ and $\lambda_3$ are the hyper-parameters.

## 2.9. Subtyping analysis

We proposed a unsupervised model with the clustering by fast search and find of density peaks (CFDP) [39] to automatically identify the depression subtypes using the subgraph embedding signatures (Fig 2C). For the clustering procedure, we first computed the Pearson correlation $J_{ij}$ between graph embeddings of every two participants. Then a distance $d_{ij}$ between every pair of participants was defined as $1 - J_{ij}$. Finally, we used CFDP method using the $d_{ij}$ as the input to obtain the depression subtypes.

Next, we describe the rules used with CFDP to select cluster centers. For each subject, CFDP computes a local density $\rho_i$ and a distance to the nearest point with higher density $\delta_i$. We plot $\delta_i$ against $\rho_i$ to obtain the decision graph and define a combined prominence score:

$$\gamma_i = \rho_i \times \delta_i. \tag{17}$$

Points with large values of both $\rho_i$ and $\delta_i$ correspond to high $\gamma_i$ and are treated as candidate cluster centers. The decision graph identifies the prominent corner region with high density and large distance, and select as centers those points with the largest $\gamma_i$. In addition, to validate the reliability of number of subtypes, we added cluster quality analyses across a range of candidate values of $K$. Using the learned graph embeddings and the corresponding CFDP assignments, we compute the silhouette coefficient and Dunn index for different choices of $K = 1 \sim 9$.

To obtain edge-level connectivity patterns of each subtype, we implemented a three-step procedure.

(1) **Computation of subject-level explanation masks.** For each subject $s$, we consider the input graph $G_s = (V_s, E_s)$ with functional connectivity edges $e \in E_s$. Using the subgraph generator module of our framework, we obtain an edge-wise explanation mask for subject $s$, that assigns to each edge a non-negative importance score

$$m_s(e) \in \mathbb{R} \geq 0, \ e \in E_s, \tag{18}$$

which quantifies how strongly edge $e$ contributes to the model's prediction for that subject.

(2) **Subject-level edge selection (top-20 edges).** Within each subject, we rank all edges in descending order of their explanation scores $m_s(e)$. Let $\{e_{s,(1)}, \ldots, e_{s,(K)}\}$ denote the $K$ edges with the largest mask values for subject $s$, i.e.,

$$m_s\left(e_{s,(1)}\right) \geq m_s\left(e_{s,(2)}\right) \geq \ldots \geq m_s\left(e_{s,(K)}\right), \tag{19}$$

with $K = 20$ in all experiments. We then define the subject-level top-$K$ edge set as

$$T_s = \{e_{s,(1)}, \ldots, e_{s,(K)}\} \subseteq E_s, \tag{20}$$

which provides a sparse, subject-specific summary of the most influential connections.

(3) **Aggregation across subjects within each subtype.** Let $c \in \{1, \ldots, C\}$ index the CFDP-derived subtypes, and

$$S_c = \{s : \textit{subject s is assigned to subtype c}\},$$

(21)

denote the set of subjects in subtype $c$. For each edge $e$ in the common edge index space, we compute a selection frequency:

$$f_c(e) = \frac{1}{|S_c|} \sum_{s \in S_c} I[e \in T_s],$$

(22)

where $I[\cdot]$ denotes the indicator function, which equals 1 if the condition is satisfied and 0 otherwise.

To assess the stability of CFDP-derived subtypes, we conducted bootstrap resampling with 1,000 iterations. In each iteration, subjects were sampled with replacement and the CFDP pipeline was re-run on the learned embeddings using identical hyperparameters, yielding a bootstrap clustering. Agreement between the reference clustering and each bootstrap clustering was quantified using the adjusted Rand index (ARI) and normalized mutual information (NMI).

We further evaluated the robustness of subtype-associated edges via 1,000 bootstrap resamples. For each resample, we recomputed the subject-level top-20 edges, aggregated them within each subtype to generate resampled subtype-level top-20 sets, and compared these sets to the corresponding reference sets using the Jaccard index.

To determine whether the observed clustering stability and edge-selection frequencies could arise from random variation, we performed a non-parametric label-permutation test (1,000 iterations). In each iteration, subtype labels were randomly permuted across subjects and the full analysis pipeline was repeated: explanation masks were re-aggregated and the edge-selection procedure was re-applied to construct null distributions of edge-selection frequencies, and the CFDP clustering was re-run to obtain null distributions for ARI and NMI. Empirical $p$-values were derived from these null distributions and adjusted for multiple comparisons using the Benjamini–Hochberg false discovery rate (FDR) procedure.

### 2.10. Brain-symptoms associations in different subtypes

Brain-symptoms associations were investigated using leave-one-out ridge regression models. On the brain side, graph embeddings for each patient were used to predict clinical symptoms. Principal component analysis (PCA) was used to reduce the dimension of graph embeddings and nine components which explained 95% of variance were retained. On the symptoms side, there were 7 significantly different clinical profiles for depression symptoms. For each ridge regression analysis, we built only one model to examine the relationship between graph embeddings and one clinical symptom score. In each loop, we optimized the regularization coefficient lambda to minimize prediction error through identifying the value of lambda between $10^{-5}$ and $10^5$. Furthermore, the performance of leave-one-subject-out ridge regression models was quantified using the coefficient of determination ($r^2$), which was defined as:

$$r^2 = 1 - \frac{\sum_i (y_i - \hat{y}_i)^2}{\sum_i (\hat{y}_i - \bar{y})},$$

(23)

where $y_i$ is the true data, $\hat{y}_i$ is prediction data and $\bar{y}$ is the average of true data. To further characterize predictive performance, we have also added the mean absolute error (MAE) as a complementary metric to $r^2$.

Permutation-based significance testing was performed for all cross-validated $r^2$ estimates. For each predictive model, we generated an empirical null distribution of $r^2$ values (1,000 iterations) by randomly permuting symptom scores and

rerunning the full analysis pipeline, while keeping the functional features, data splits and model hyperparameters identical to those used in the original analysis. The observed cross-validated $r^2$ was then compared against this null distribution to derive an empirical permutation $p$ value. To account for multiple testing across symptom dimensions, $p$ values were adjusted using false discovery rate (FDR) correction.

## 2.11. Baselines

For comparison, we evaluate the performance of our EH-BrainGNN against four traditional psychiatric diagnostic classifiers including support vector machines (SVM) with linear and RBF kernel [40], LASSO [41] and random forest (RF) [42]; three baseline GNNs including GCN [43], GAT [44] and GIN [32]; two state-of-the-art (SOTA) GNN explainers including DIR-GNN and ProtGNN [45]; and three GNNs designed for brain networks including BrainGNN [46], CI-GNN [20] and BrainIB [21].

## 2.12. Training, testing and hyperparameter optimization

We evaluated model performance using cross-validation [47] and independent external validation, and report the area under the receiver operating characteristic curve (AUC) and accuracy across folds. For cross-national validation, to provide a balanced assessment of classification performance, we additionally report the F1 score and sensitivity alongside balanced accuracy. For primary experiments, we adopted an outer 10-fold cross-validation scheme to split the dataset into training and held-out test folds. Within each outer fold, all preprocessing and model fitting steps were performed using only the outer training data, and the resulting model was then evaluated on the corresponding held-out fold.

Hyperparameters were optimized using a nested cross-validation procedure to ensure unbiased performance estimation. Specifically, within each outer training split, we conducted an inner 5-fold cross-validation to select hyperparameters. For each candidate configuration defined in Table 2 (including learning rate, batch size, number of layers, hidden units, weight decay, dropout rate and other model-specific settings), the model was trained on four inner folds and evaluated on the remaining fold, iterating over all inner folds. The configuration that maximized mean inner-fold accuracy was selected. Using this fixed configuration, the model was then retrained on the full outer training set and evaluated on the outer test fold. Early stopping was applied with a patience of 10 epochs, monitoring validation accuracy on an internal validation split drawn exclusively from the outer training data. At no stage were outer test folds or the independent validation cohort used for hyperparameter tuning, model selection or early-stopping calibration.

**Table 2. Range of hyper-parameters and final specification for EH-BrainGNN.**

| Hyper-parameter | Range Examined | Final Specification (10-fold cross validation) | Final Specification (leave-one-site-out cross validation) | Final Specification (cross-national validation) |
|---|---|---|---|---|
| Epoch | [350] | 350 | 350 | 350 |
| #GNN Layers | [2,3,4,5,6] | 5 | 5 | 5 |
| #MLP Layers | [2,3,4] | 2 | 2 | 2 |
| #Hidden Dimensions | [64,128,256,512] | 128 | 128 | 128 |
| Learning Rate | [1e-2,1e-3,1e-4] | 1e-3 | 1e-3 | 1e-3 |
| Batch Size | [32,64,128] | 64 | 64 | 64 |
| Weight Decay | [5e-3,5e-4] | 5e-3 | 5e-4 | 5e-3 |
| Dropout | [0.5] | 0.5 | 0.5 | 0.5 |
| $\tau$ | [0.1,1] | 1 | 1 | 1 |
| $\lambda_1$ | [1e-2,1e-3,1e-4,1e-5] | 1e-4 | 1e-4 | 1e-3 |
| $\lambda_2$ | [1e-2,1e-3,1e-4,1e-5] | 1e-5 | 1e-4, | 1e-4, |
| $\lambda_3$ | [1e-2,1e-3,1e-4,1e-5] | 1e-4 | 1e-3 | 1e-4 |

Model training used the Adam optimizer with a learning rate of 0.001, a dropout rate of 0.5 and weight decay of 0.0005, unless otherwise specified. The GIN architecture comprised five layers and was trained with a batch size of 64. For EH-BrainGNN, hyperparameters were selected either by grid search within the nested cross-validation framework or set according to recommended values from prior work; the search ranges and final specifications are reported in Table 2.

To prevent information leakage during site harmonization, ComBat parameters were always estimated exclusively from training data within each evaluation split and then applied to held-out data without refitting. For the 10-fold cross-validation experiments, ComBat was fitted on the outer training data within each fold and applied to the corresponding held-out fold. For leave-one-site-out (LOSO) cross-validation, ComBat was fitted using data from the training sites only and then applied to the left-out site; no information from the held-out site, including its distributional characteristics, was used when fitting the harmonization model. For independent cross-national validation, ComBat was fitted solely on the Chinese training sites, and the learned parameters were subsequently applied to the Japanese cohort, which was treated as a strictly unseen target domain; no ComBat parameters were estimated using any Japanese labels or Japanese distributional information.

## 3. Results

### 3.1. Generalization performance: Ten-fold cross-validation

To assess the performance, we conducted 10-fold cross-validation procedure [47] in both datasets. Furthermore, we used ComBat harmonization method [25–28] to mitigate distinct site differences and artifact or confounding effect. Table 3 demonstrate the classification performance for different models. Using 10-fold cross-validation, EH-BrainGNN achieved an accuracy of 74% in the principal dataset and an accuracy of 73% in the independent validation dataset. Compared with other methods, EH-BrainGNN yielded significant and consistent improvements in both datasets, suggesting that the model had formidable discriminatory ability. To assess the statistical significance of performance differences, we compared EH-BrainGNN against the next-best method using paired tests across the 10 cross-validation folds (fold-wise accuracy as paired observations). The results are shown in Fig 4 A-B. For the principal dataset, EH-BrainGNN was compared to BrainIB in terms of accuracy: $t=4.3529$, $p=0.0018$, Cohen's $d=1.3765$, with a bootstrap 95% confidence interval (CI) of [0.7998, 2.9158]. For the independent validation dataset, EH-BrainGNN was compared to CI-GNN (accuracy): $t=3.1623$, $p=0.0115$, Cohen's $d=1.0000$, with a bootstrap 95% CI of [0.4454, 2.4244]. In addition to the high classification

**Table 3. Tenfold cross validation performance on different models in both datasets. The highest performance is highlighted with boldface.**

| Datasets Methods | Principal Dataset | | Independent validation dataset | |
|---|---|---|---|---|
| | ACC | AUC | ACC | AUC |
| LASSO | 0.62 (0.02) | 0.62 (0.02) | 0.61 (0.06) | 0.59 (0.05) |
| RF | 0.67 (0.02) | 0.67 (0.02) | 0.60 (0.03) | 0.64 (0.13) |
| L-SVM | 0.65 (0.02) | 0.65 (0.02) | 0.56 (0.01) | 0.54 (0.02) |
| RBF-SVM | 0.69 (0.01) | 0.69 (0.01) | 0.62 (0.05) | 0.57 (0.07) |
| GCN | 0.61 (0.01) | 0.60 (0.02) | 0.68 (0.04) | 0.65 (0.06) |
| GAT | 0.65 (0.02) | 0.64 (0.01) | 0.64 (0.08) | 0.65 (0.06) |
| GIN | 0.65 (0.03) | 0.64 (0.02) | 0.57 (0.10) | 0.57 (0.11) |
| DIR-GNN | 0.64 (0.02) | 0.63 (0.01) | 0.68 (0.03) | **0.68 (0.05)** |
| ProtGNN | 0.61 (0.02) | 0.60 (0.01) | 0.67 (0.02) | 0.67 (0.03) |
| BrainGNN | 0.60 (0.03) | 0.60 (0.01) | 0.67 (0.08) | 0.66 (0.06) |
| CI-GNN | 0.67 (0.04) | 0.67 (0.01) | 0.70 (0.04) | 0.67 (0.04) |
| BrainIB | 0.70 (0.02) | 0.69 (0.01) | 0.68 (0.05) | 0.66 (0.06) |
| EH-BrainGNN | **0.74 (0.02)** | **0.73 (0.02)** | **0.73 (0.01)** | **0.68 (0.04)** |

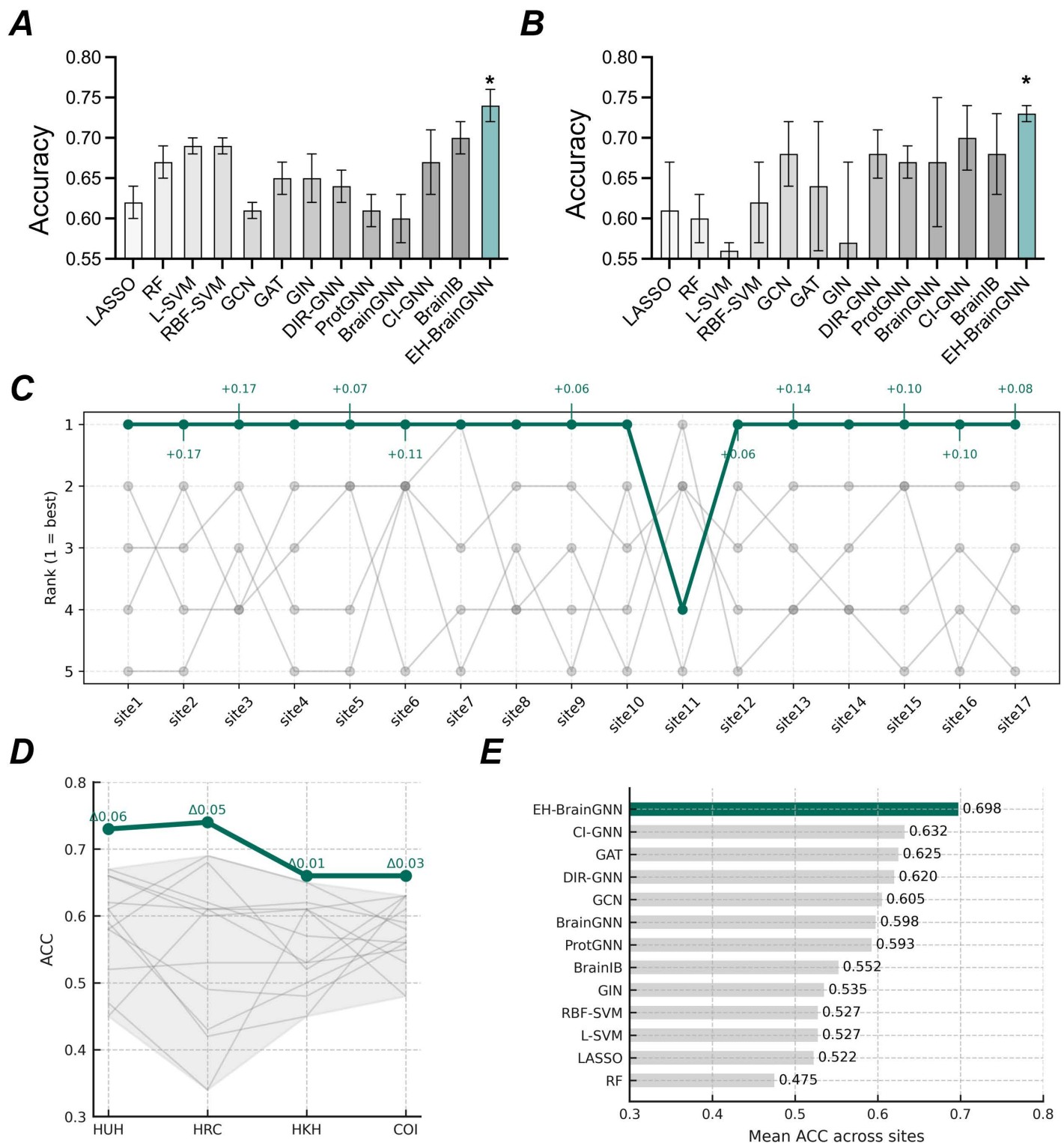

**Fig 4. Generalization performance of EH-BrainGNN. (A)** Tenfold cross validation performance on different models in principal dataset. **(B)** Tenfold cross validation performance on different models in replication dataset. **(C)** The leave-one-site-out cross-validation performance ranking of different models based on their accuracy. **(D-E)** Generalization Performance of the classifier for independent cohorts.

performance of EH-BrainGNN, the framework provided an information-condensed signature (graph embeddings) which yielded meaningful binary discrimination between depressed patients and healthy controls (S2 Fig A-B).

### 3.2. Generalization Performance: Leave-one site out cross-validation

To assess the generalizability of classification model to unseen data collected at completely different sites, a leave-one-site-out analysis was performed by dividing the dataset into the training set (16 sites out of 17 sites) to train the model, and the testing set (the left site out of 17 sites) for testing a model (S3 Fig) in the principal dataset. A ComBat harmonization method [25–28] for training set was applied to mitigate distinct site differences and artifact or confounding effect. Then we fitted a model to each site-sample resulting 17 MDD classifiers. We then calculated the area under the curve (AUC) and accuracy for every classifier which were shown in Table 4. And we evaluated the leave-one-site-out cross-validation performance ranking of different models based on their accuracy (Fig 4C).

As can be seen, EH-BrainGNN achieved the highest average generalization accuracy across sites (73%) compared with the baseline models. Improvements were broadly consistent across sites, indicating better generalization under site-specific variability. Moreover, from the performance rankings across different models, our model consistently maintained the top position across most sites. Although it ranked fourth at site 11, the accuracy reached 0.84, which is still competitive, with the highest accuracy across models being 0.88. This discrepancy may be attributed to the relatively older average age of participants at this site, where the mean age of MDD patients is 46 years.

**Table 4. Leave-one-site-out cross validation performance on different models in principal dataset. The highest performance is highlighted with boldface.**

| Method Sites | LASSO | | RF | | L-SVM | | RBF-SVM | | EH-BrainGNN | |
|---|---|---|---|---|---|---|---|---|---|---|
| | ACC | AUC | ACC | AUC | ACC | AUC | ACC | AUC | ACC | AUC |
| site1 | 0.56 | 0.56 | 0.48 | 0.48 | 0.62 | 0.62 | 0.54 | 0.54 | **0.63** | **0.63** |
| site2 | 0.53 | 0.54 | 0.40 | 0.40 | 0.43 | 0.44 | 0.63 | 0.65 | **0.80** | **0.79** |
| site3 | 0.71 | 0.74 | 0.68 | 0.69 | 0.63 | 0.67 | 0.63 | 0.66 | **0.88** | **0.89** |
| site4 | 0.67 | 0.69 | 0.64 | 0.64 | 0.72 | 0.73 | 0.74 | **0.75** | 0.75 | 0.75 |
| site5 | 0.61 | 0.60 | 0.60 | 0.62 | 0.63 | 0.63 | 0.63 | 0.64 | **0.70** | **0.70** |
| site6 | 0.58 | 0.58 | 0.58 | 0.60 | 0.57 | 0.58 | 0.58 | 0.58 | **0.69** | **0.69** |
| site7 | 0.65 | 0.64 | **0.75** | **0.73** | 0.68 | 0.65 | 0.72 | 0.70 | **0.75** | 0.70 |
| site8 | 0.65 | 0.68 | 0.54 | 0.53 | 0.54 | 0.54 | 0.68 | **0.70** | **0.70** | **0.70** |
| site9 | 0.58 | 0.58 | 0.69 | 0.69 | 0.64 | 0.67 | 0.72 | 0.72 | **0.78** | **0.78** |
| site10 | 0.70 | 0.66 | 0.57 | 0.52 | 0.61 | 0.59 | 0.68 | 0.63 | **0.72** | **0.68** |
| site11 | 0.78 | 0.77 | 0.87 | 0.86 | **0.88** | **0.90** | 0.87 | 0.87 | 0.84 | 0.82 |
| site12 | 0.61 | 0.61 | 0.55 | 0.55 | 0.51 | 0.51 | 0.56 | 0.56 | **0.67** | **0.67** |
| site13 | 0.51 | 0.56 | 0.45 | 0.54 | 0.45 | 0.47 | 0.59 | 0.67 | **0.73** | **0.73** |
| site14 | 0.55 | 0.55 | 0.57 | 0.57 | 0.55 | 0.55 | 0.61 | 0.61 | **0.63** | **0.62** |
| site15 | 0.56 | 0.56 | 0.58 | 0.58 | 0.55 | 0.55 | 0.58 | 0.59 | **0.68** | **0.67** |
| site16 | 0.61 | 0.62 | 0.53 | 0.54 | 0.58 | 0.60 | 0.66 | 0.68 | **0.76** | **0.76** |
| site17 | 0.47 | 0.47 | 0.49 | 0.49 | 0.40 | 0.39 | 0.56 | 0.56 | **0.64** | **0.65** |
| Mean | 0.61 | 0.61 | 0.59 | 0.59 | 0.59 | 0.59 | 0.65 | 0.65 | **0.73** | **0.72** |

ACC = Accuracy; AUC = Area Under Curve; RF = random forest; L-SVM = SVM with linear kernel; RBF-SVM = SVM with radial basis function kernel.

### 3.3. Generalization Performance: Generalization of the classifier for independent cohorts

To further test the generalizability of EH-BrainGNN, we used Chinese-population-based data to construct classifier to predict Japanese-population-based data. To be specific, we used data from the principal dataset to train the model, and applied trained classifier to the independent validation dataset (data for each site). Table 5 and Fig 4D-E demonstrate generalization performance in HUH, HRC, HKH and COI from the independent validation dataset. EH-BrainGNN demonstrated a generalization accuracy of 73% and an AUC of 72% at HUH, with an absolute improvement of over 16% in accuracy compared to RBF-SVM. More importantly, EH-BrainGNN achieved the highest performance in terms of multiple metrics, including accuracy, sensitivity, and F1-score, across most sites (Table 5). Specifically, in HRC, HKH, and COI, EH-BrainGNN outperformed all baseline models, showing significant improvements in both accuracy and AUC. At HUH, the model achieved an accuracy of 73%, sensitivity of 79%, and F1-score of 76%. At HRC, the accuracy was 74%, sensitivity 86%, and F1-score 83%. For HKH, EH-BrainGNN yielded an accuracy of 66%, an AUC of 67%, and a sensitivity of 72%, while in COI, the model achieved a balanced accuracy of 66%, an AUC of 63%, and sensitivity of 75%. In contrast, other models like RBF-SVM, CI-GNN and BrainIB showed lower generalization performance across these sites, with accuracy values mostly below 70%, and poor sensitivity and F1-scores in certain cases. These results suggested that a reliable classifier that we developed by only using the training data obtained from China, could be successfully applied to classify MDD and HCs in the Japan validation cohorts. Generalization performance across different models on the independent validation dataset, trained on the principal dataset. The best-performing result is highlighted in bold (Fig 5).

### 3.4. Subtyping Analysis: Graph embeddings define three depression subtypes

According to our overall frameworks, we further adopted unsupervised learning to these high-dimensional signatures obtained from deep-learning model to identify subtypes of MDD. To ensure that cluster discovery was not confounded by the data distribution and the number of clusters, we employed a data-driven method (clustering by fast search and find of density peaks, CFDP) which automatically identified the number of clusters and MDD subtypes. The decision graph and subtype distribution are shown in Fig 6A, C. Three depression subtypes were identified, comprising 356 (43.0%),

**Table 5. Generalization performance on different models in the independent validation dataset using the principal dataset. The highest performance is highlighted with boldface.**

| Sites Method | HUH | | | | HRC | | | | HKH | | | | COI | | | |
|---|---|---|---|---|---|---|---|---|---|---|---|---|---|---|---|---|
| | ACC | AUC | Sen | F1 | ACC | AUC | Sen | F1 | ACC | AUC | Sen | F1 | ACC | AUC | Sen | F1 |
| LASSO | 0.59 | 0.60 | 0.51 | 0.58 | 0.43 | 0.58 | 0.27 | 0.41 | 0.50 | 0.50 | 0.48 | 0.48 | 0.57 | 0.56 | 0.66 | 0.66 |
| RF | 0.47 | 0.54 | 0.41 | 0.47 | 0.34 | 0.46 | 0.16 | 0.27 | 0.61 | 0.61 | 0.55 | 0.58 | 0.48 | 0.56 | 0.41 | 0.50 |
| L-SVM | 0.58 | 0.59 | 0.74 | 0.67 | 0.49 | 0.53 | 0.39 | 0.52 | 0.48 | 0.49 | 0.24 | 0.30 | 0.56 | 0.59 | 0.65 | 0.65 |
| RBF-SVM | 0.61 | 0.61 | 0.59 | 0.62 | 0.42 | 0.55 | 0.35 | 0.47 | 0.45 | 0.46 | 0.21 | 0.27 | 0.63 | **0.63** | 0.67 | 0.71 |
| GCN | 0.67 | 0.68 | 0.69 | 0.70 | 0.62 | 0.52 | 0.69 | 0.72 | 0.57 | 0.58 | 0.52 | 0.55 | 0.56 | 0.53 | 0.62 | 0.64 |
| GAT | 0.66 | 0.67 | 0.59 | 0.66 | 0.60 | 0.58 | 0.65 | 0.71 | 0.61 | 0.61 | 0.34 | 0.45 | 0.63 | 0.58 | 0.70 | 0.71 |
| GIN | 0.45 | 0.48 | 0.03 | 0.05 | 0.61 | 0.61 | 0.59 | 0.70 | 0.53 | 0.53 | 0.66 | 0.55 | 0.55 | 0.52 | 0.62 | 0.64 |
| DIR-GNN | 0.66 | 0.67 | 0.75 | 0.71 | 0.61 | 0.61 | 0.59 | 0.68 | 0.62 | 0.64 | 0.55 | 0.59 | 0.59 | 0.59 | 0.56 | 0.62 |
| ProtGNN | 0.62 | 0.62 | 0.56 | 0.62 | 0.61 | 0.59 | 0.65 | 0.72 | 0.61 | 0.61 | 0.55 | 0.58 | 0.53 | 0.55 | 0.49 | 0.57 |
| BrainGNN | 0.58 | 0.59 | 0.63 | 0.63 | 0.68 | 0.57 | 0.78 | 0.76 | 0.52 | 0.53 | 0.48 | 0.49 | 0.61 | 0.50 | 0.64 | 0.67 |
| CI-GNN | 0.61 | 0.61 | 0.62 | 0.63 | 0.69 | 0.58 | 0.78 | 0.77 | 0.65 | 0.65 | 0.62 | 0.62 | 0.58 | 0.57 | 0.58 | 0.63 |
| BrainIB | 0.52 | 0.71 | 0.39 | 0.64 | 0.53 | 0.63 | 0.59 | 0.70 | 0.53 | 0.52 | 0.52 | 0.52 | 0.63 | 0.58 | **0.86** | **0.74** |
| EH-BrainGNN | **0.73** | **0.72** | **0.79** | **0.76** | **0.74** | **0.62** | **0.86** | **0.83** | **0.66** | **0.67** | **0.72** | **0.67** | **0.66** | **0.63** | 0.75 | **0.74** |

ACC = Accuracy; AUC = Area Under Curve; Sen = Sensitivity; RF = random forest; L-SVM = SVM with linear kernel; RBF-SVM = SVM with radial basis function kernel.

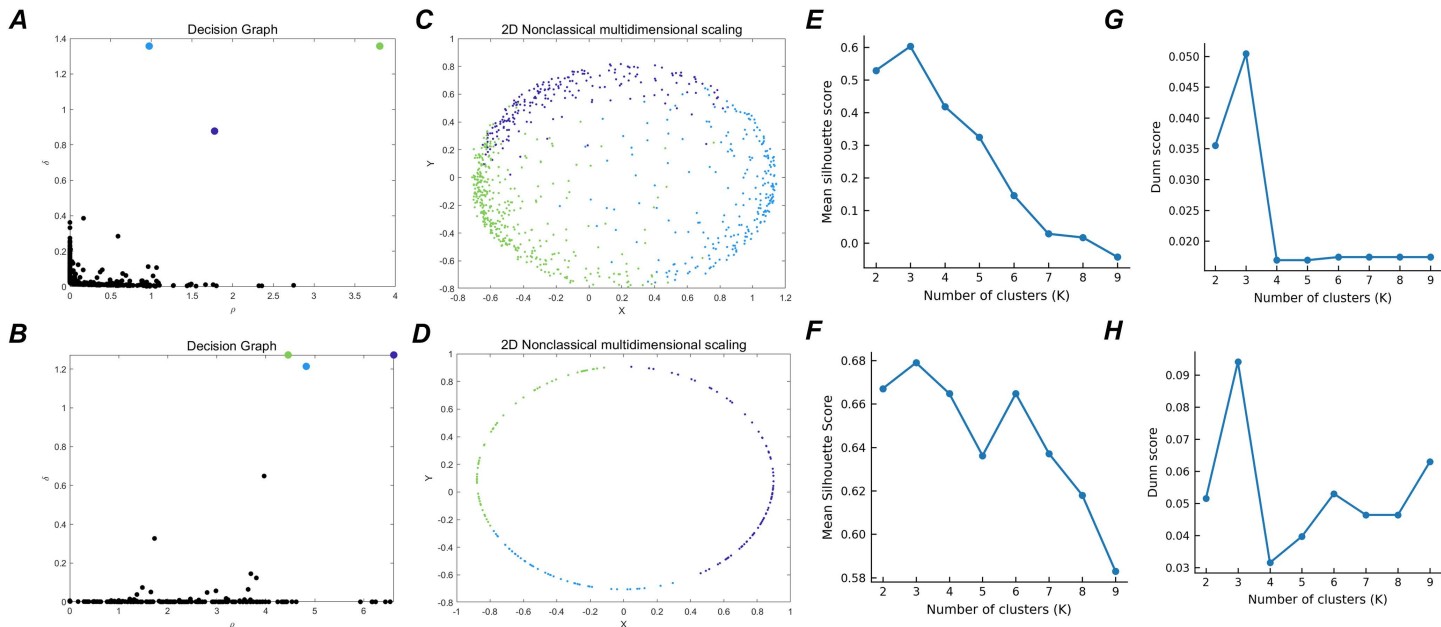

**Fig 5. Determination of the optimal number of CFDP subtypes. (A-B)** CFDP decision graphs for the two datasets. **(C-D)** Two dimensional embeddings illustrating the spatial distribution of subjects colored by CFDP subtype. **(E-F)** Silhouette coefficients as a function of the number of clusters ($K = 2-9$). **(G-H)** Dunn indices for $K = 2-9$.

226 (27.3%) and 246 (29.7%) patients, respectively. To assess clustering robustness, we computed the silhouette coefficient and Dunn index across a range of candidate subtype numbers (Fig 6E, G). Both indices consistently supported a three-subtype solution.

To ensure that subtypes were biologically meaningful, we compared HAMD total scores and each item of HAMD using two-sided Wilcoxon rank-sum tests. Multiple comparisons across items were controlled using the Benjamini–Hochberg false discovery rate (FDR) procedure, with significance assessed at a preset threshold of $q < 0.05$. We observed that HAMD total scores were significantly different between subtype1 and subtype3 ($p = 0.029$, $d = 0.41$, 95%CI = [0.04, 0.77], Fig 7B). In addition, seven symptom items of HAMD showed significant differences (Fig 7A) including depressed mood (subtype2 vs. subtype3: $p = 0.027$, $d = 0.29$, 95% CI =[0.11, 0.47]), insomnia-early (subtype1 vs. subtype3: $p = 0.010$, $d = 0.61$, 95% CI = [0.45, 0.78]; subtype2 vs. subtype3: $p = 0.018$, $d = 0.43$, 95% CI =[0.24, 0.61]), insomnia-middle (subtype2 vs. subtype3: $p = 0.007$, $d = 0.31$, 95% CI = [0.13, 0.50]), agitation (subtype2 vs. subtype3: $p = 0.026$, $d = 0.32$, 95% CI = [0.13, 0.50]), genital (subtype2 vs. subtype3: $p = 0.026$, $d = 0.47$, 95% CI = [0.28, 0.65]), weight loss (subtype2 vs. subtype3: $p = 0.048$, $d = 0.24$, 95% CI = [0.06, 0.42]) and insight (subtype2 vs. subtype3: $p = 0.026$, $d = 0.30$, 95% CI = [0.12, 0.48]).

Furthermore, we further validated the consistency and reproducibility of our identified MDD subtypes in the independent validation dataset. Interestingly, we consistently identified three MDD subtypes in the independent validation dataset (Fig 6B, D, F, H). Three depression subtypes were identified, comprising 43 (24.3%), 60 (33.9%) and 74 (41.8%) patients, respectively. A trend towards a reduced pattern of BDI scores was observed between subtype 2 and subtype 3 (BDI scores of subtype 2 = 27.21 ± 10.69, and BDI scores of subtype 3 = 30.05 ± 9.79, respectively; $p = 0.066$, $d = -0.28$, 95% CI = [-0.67, 0.10]), but no statistically significant difference was found (Fig 7C).

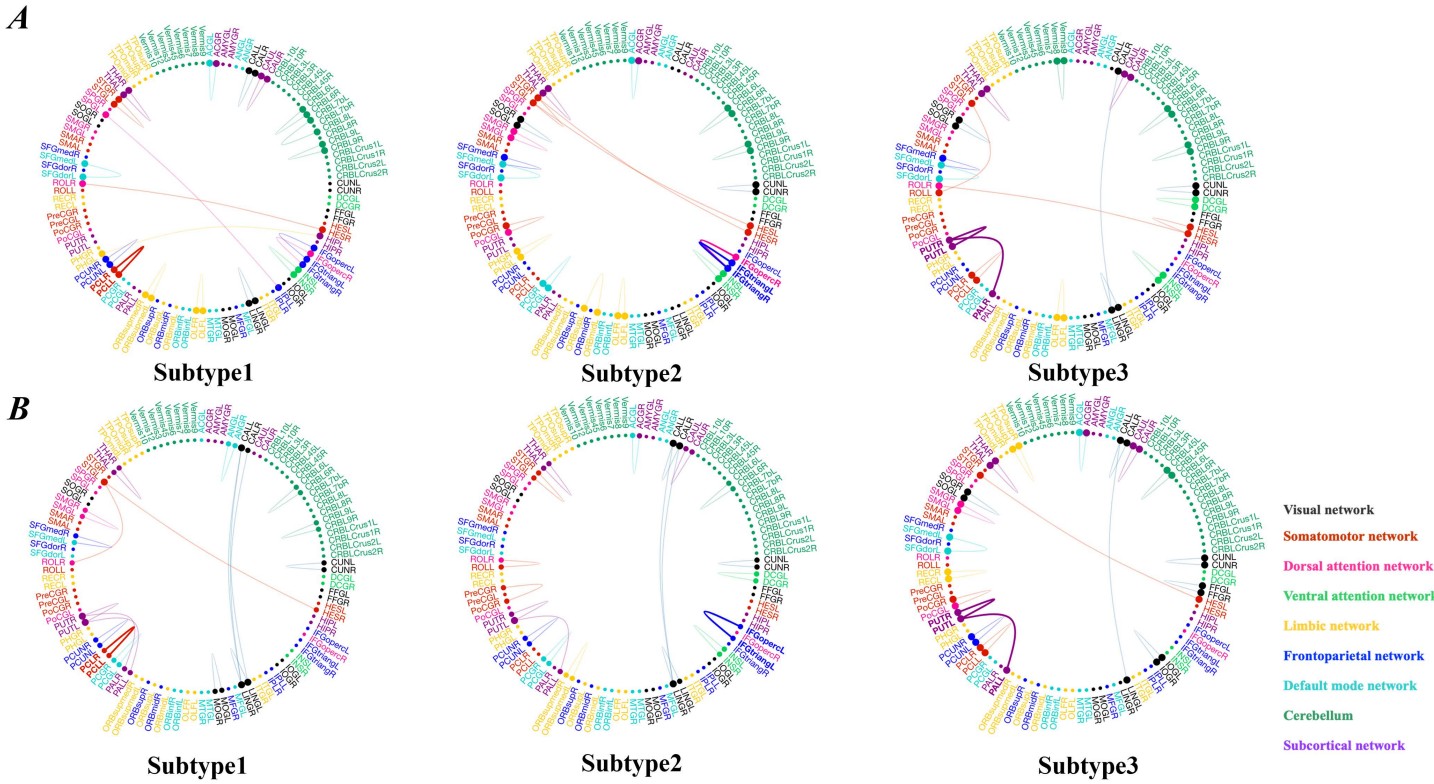

**Fig 6. Visualization of distinct patterns in different subtypes of MDD in principal dataset and independent validation dataset, respectively.** The colors of brain neural systems are described as: visual network (VN), somatomotor network (SMN), dorsal attention network (DAN), ventral attention network (VAN), limbic network (LIN), frontoparietal network (FPN), default mode network (DMN), cerebellum (CBL) and subcortical network (SBN), respectively.

### 3.5. Subtyping Analysis: Subtype-specific brain connections

To illustrate specific brain patterns in each subtype of MDD, we compared subgraphs obtained from explanation generator among three subtypes (Fig 7). The consistent subtype-specific brain connections on both datasets are summarized in Fig 8d and Table D in S1 Text, and we have the following observations:

(1) We observed that connection between left and right thalamus overlapped among three subtypes, suggesting that this connection played an essential role in MDD diagnosis no matter subtype of MDD. This result is consistent with a recent study [48], where they found that MDD classifications were driven by stronger thalamic connections and the connection between the left and right thalamus is one of the most important connection for classification MDD vs HC.

(2) Common diagnostic features existed among different subtypes of major depression and identified 3 common connections between a) left and right cerebellum.6 between subtype1 and subtype2; b) left and right paracentral lobule between subtype1 and subtype3; c) left and right cuneus between subtype2 and subtype3.

(3) Different subtypes of MDD also showed distinct brain patterns. Specifically, MDD subtype 1 was characterized by connections within the DMN (precuneus)-Cerebellum network which were associated with abnormal insomnia-early. Previous study has reported that insomnia is associated with impaired disengagement of brain regions involved in self-referential processes (precuneus) [49]. Brain patterns of MDD patients of subtype 2 mainly encompassed connections within Insula-Cingulum-Temporal network, where connections between left and right insula which was involved

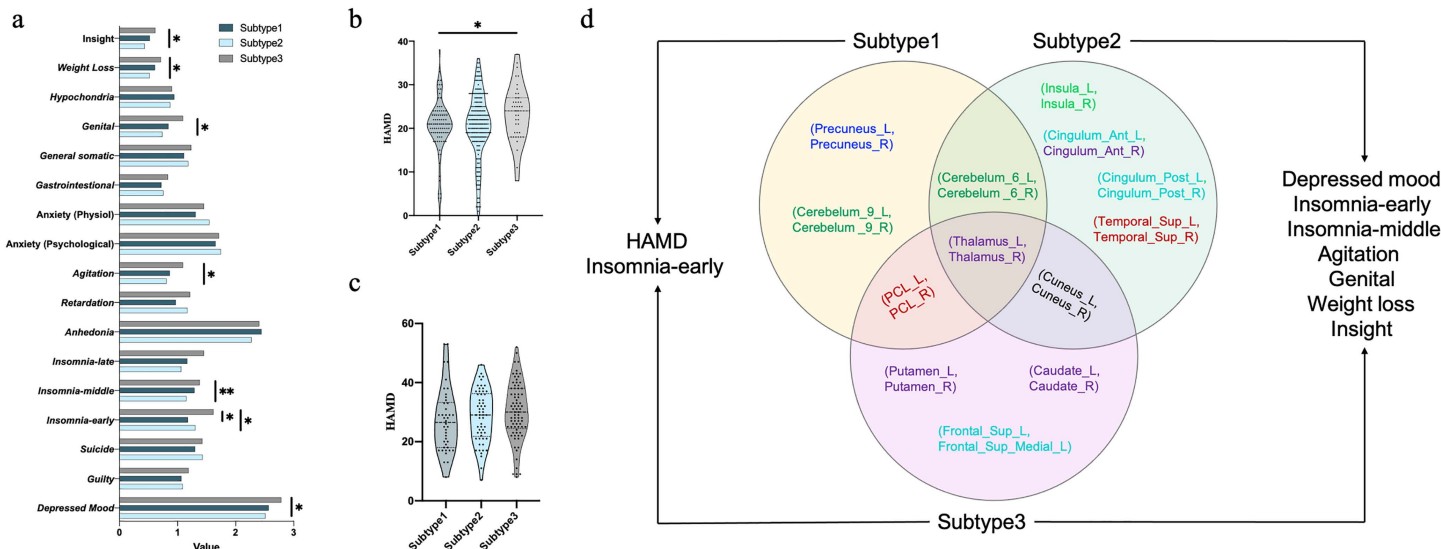

**Fig 7. Graph embeddings define depression subtypes. (a)** Comparison of clinical profiles for depression symptoms (HAMD-17) among three depression subtypes using Wilcoxon rank-sum tests with FDR correction on the principal dataset. **(b)** Comparison of HAMD total scores among three depression subtypes using Wilcoxon rank-sum tests with FDR correction on the principal dataset. **(c)** Comparison of BDI-II scores among three depression subtypes using Wilcoxon rank-sum tests on the independent validation dataset. **(d)** The overlap and distinction of brain patterns and symptom profiles within three subtypes in principal dataset and independent validation dataset.

in socio-emotional processing and general attention [50], were associated with reduced depressed mood and insight in subtype 2. MDD subtype 3 was characterized by connections within the frontostriatal network involved in reward processing and action initiation [51–53], including connections between 1) left and right putamen; 2) left and right caudate; 3) left superior frontal cortex and left superior medial frontal cortex. These connections were associated with increased insomnia, agitation and weight loss.

### 3.6. Subtyping Analysis: Stability and significance of subtype-specific patterns

To assess the stability of CFDP-derived subtypes, we performed bootstrap resampling (1,000 iterations) and quantified agreement with the original solution using the adjusted Rand index (ARI) and normalized mutual information (NMI). Subtype assignments showed substantial reproducibility (ARI: mean = 0.523, 95% CI = [0.136, 0.863]; NMI: mean = 0.543, 95% CI = [0.198, 0.809]). To determine whether this stability exceeded chance, we compared the observed stability metrics with a null distribution generated from random cluster assignments matched for subtype number and size. The observed distributions were markedly shifted relative to null, yielding large effect sizes (Cohen's $d$ for ARI = 3.13; Cohen's $d$ for NMI = 4.38), indicating that CFDP-derived subtypes were substantially more reproducible than expected under random clustering.

Subtype-associated edge sets showed moderate-to-high stability across bootstrap resamples (S4 Fig A). The mean Jaccard overlap was 0.433 (95% CI [0.250, 0.600]) for subtype 1, 0.405 (95% CI [0.176, 0.667]) for subtype 2 and 0.335 (95% CI [0.143, 0.538]) for subtype 3. As both the reference and bootstrap-derived sets comprised 20 edges, these overlaps correspond to ~10–12 shared edges on average, indicating that more than half of the reported edges were consistently recovered across resamples, whereas the remainder formed a more variable peripheral set. The overlaps were well above chance (random top-20 selection yields Jaccard values close to zero), supporting a reproducible core of subtype-specific connections. Consistently, many reference edges showed high bootstrap selection frequencies (> 0.5): 19/20 edges for subtype 1, 16/20 for subtype 2 and 9/20 for subtype 3 (S4 Fig B-D).

Permutation testing further indicated that both the subtype-specific top-20 edge patterns and the CFDP clustering stability metrics were highly unlikely under a random-label null model (empirical $p < 0.0001$ across 1,000 permutations).

### 3.7. Correspondences between functional signature and MDD symptoms

To further investigate the association between patterns of discriminative functional signature and clinical symptoms, we used graph embeddings of subgraph and significantly different clinical profiles for depression symptoms (HAMD) to generate GE-HAMD models using leave-one-subject-out ridge regression. We present the $r^2$ (95% CI) and MAE for each subtype in Table 6, along with scatter plots showing predicted versus observed symptom scores in S5 Fig. Our findings indicate that graph embeddings serve as strong predictors of clinical indicators across all subtypes, with R² values greater than 0.3 for each subtype. Additionally, symptom profiles in subtype 3 were better predicted by graph embeddings, which aligns with the significantly enhanced symptom profiles observed in subtype 3. To further assess the robustness of our results, we performed a permutation test (Table 6), which showed that, after multiple comparison correction, the *p*-values were less than 0.0001.

### 3.8. Validation analysis

We validated our results by considering several potential confounding factors.

(1) we added an additional analysis to investigate the extent of head motion contribution to predictability with/without age gender effects. Specifically, we conducted a ten-fold cross validation under four different settings in the principal dataset. We observed the performances of classifiers did not show any significant changes under different settings. These results suggested that it is the neural basis rather than the artifact or confounding that contribute to classification (Table 7).

(2) we have added an additional experiment to investigate the effect of ROI selection. Specifically, we repeated the entire 10-fold cross-validation procedure using the most popular brain atlas (Power Atlas 49) including 264 brain regions. We observed that model based on Power atlas outperformed model based on AAL atlas for all approaches except random forest, suggesting that model performance was positively correlated with the number of ROIs (Table 8).

(3) we repeated leave-one-site-out analysis without harmonization and compared prediction performances to assess effects of Combat harmonization method. We found significant improvements in the prediction performances using harmonization for some sites (Table 9). To ensure that harmonization did not remove disease-relevant effects, we performed a case–control diagnostic comparing MDD versus HC functional connectivity before and after ComBat. As

Table 6. Subtype-specific clinical profiles for significantly different depression symptoms. The highest performance is highlighted with boldface.

| Symptoms | Subtype1 | | | | Subtype2 | | | | Subtype3 | | | |
|---|---|---|---|---|---|---|---|---|---|---|---|---|
| | $R^2$ | 95% CI | MAE | p values | $R^2$ | 95% CI | MAE | p values | $R^2$ | 95% CI | MAE | p values |
| Depressed Mood | 0.48 | [0.34, 0.60] | 0.55 | $< 10^{-4}$ | 0.34 | [0.25, 0.43] | 0.65 | $< 10^{-4}$ | **0.56** | [0.42, 0.71] | **0.39** | $< 10^{-4}$ |
| Insomnia early | 0.46 | [0.33, 0.55] | 0.46 | $< 10^{-4}$ | 0.37 | [0.28, 0.43] | 0.53 | $< 10^{-4}$ | **0.56** | [0.28, 0.72] | **0.33** | $< 10^{-4}$ |
| Insomnia middle | 0.48 | [0.35, 0.58] | 0.42 | $< 10^{-4}$ | 0.35 | [0.26, 0.42] | 0.47 | $< 10^{-4}$ | **0.54** | [0.30, 0.73] | **0.40** | $< 10^{-4}$ |
| Agitation | 0.45 | [0.30, 0.54] | **0.47** | $< 10^{-4}$ | 0.42 | [0.31, 0.51] | 0.52 | $< 10^{-4}$ | **0.51** | [0.31, 0.70] | **0.47** | $< 10^{-4}$ |
| Genital | 0.53 | [0.43, 0.61] | 0.43 | $< 10^{-4}$ | 0.44 | [0.37, 0.52] | 0.47 | $< 10^{-4}$ | **0.57** | [0.41, 0.68] | **0.40** | $< 10^{-4}$ |
| Weight Loss | 0.40 | [0.29, 0.48] | **0.47** | $< 10^{-4}$ | 0.34 | [0.25, 0.42] | 0.48 | $< 10^{-4}$ | **0.57** | [0.43, 0.68] | **0.47** | $< 10^{-4}$ |
| Insight | 0.44 | [0.31, 0.59] | 0.37 | $< 10^{-4}$ | 0.36 | [0.29, 0.43] | 0.39 | $< 10^{-4}$ | **0.56** | [0.43, 0.63] | **0.34** | $< 10^{-4}$ |

**Table 7. The effects of age, gender, head motion on ten-fold cross validation performance on different models in the principal dataset.**

| Groups Methods | Group with age, gender effects | | Group without age, gender effects | |
|---|---|---|---|---|
| | With head motion | Without head motion | With head motion | Without head motion |
| LASSO | 0.62 (0.02) | 0.61 (0.01) | 0.60 (0.02) | 0.61 (0.01) |
| RF | 0.67 (0.02) | 0.66 (0.01) | 0.66 (0.01) | 0.64 (0.01) |
| L-SVM | 0.65 (0.02) | 0.64 (0.02) | 0.64 (0.02) | 0.65 (0.02) |
| RBF-SVM | 0.69 (0.01) | 0.69 (0.01) | 0.69 (0.01) | 0.69 (0.01) |
| EH-BrainGNN | **0.74 (0.02)** | **0.73 (0.02)** | **0.72 (0.01)** | **0.72 (0.01)** |

RF = random forest; L-SVM = SVM with linear kernel; RBF-SVM = SVM with radial basis function kernel.

**Table 8. The effects of brain atlas on ten-fold cross validation performance on different models in the principal dataset.**

| Atlases Methods | AAL | | Power Atlas | |
|---|---|---|---|---|
| | ACC | AUC | ACC | AUC |
| LASSO | 0.62 (0.02) | 0.61 (0.01) | 0.65 (0.02) | 0.65 (0.02) |
| RF | 0.67 (0.02) | 0.66 (0.01) | 0.64 (0.02) | 0.64 (0.03) |
| L-SVM | 0.65 (0.02) | 0.64 (0.02) | 0.64 (0.03) | 0.65 (0.04) |
| RBF-SVM | 0.69 (0.01) | 0.69 (0.01) | 0.69 (0.01) | 0.69 (0.02) |
| EH-BrainGNN | **0.74 (0.02)** | **0.73 (0.02)** | **0.75 (0.02)** | **0.76 (0.01)** |

ACC = Accuracy; AUC = Area Under Curve; RF = random forest; L-SVM = SVM with linear kernel; RBF-SVM = SVM with radial basis function kernel.

**Table 9. Effects of harmonization for the leave-one-site-out analysis.**

| Sites | ACC (Before Combat) | ACC (After Combat) | △ ACC | AUC (Before Combat) | AUC (After Combat) | △ AUC |
|---|---|---|---|---|---|---|
| Site1 | 0.54 | 0.63 | 0.09 | 0.54 | 0.63 | 0.09 |
| Site2 | 0.67 | 0.80 | 0.13 | 0.66 | 0.79 | 0.13 |
| Site3 | 0.85 | 0.88 | 0.03 | 0.86 | 0.89 | 0.03 |
| Site4 | 0.79 | 0.75 | -0.04 | 0.79 | 0.75 | -0.04 |
| Site5 | 0.71 | 0.70 | -0.01 | 0.73 | 0.70 | -0.03 |
| Site6 | 0.64 | 0.69 | 0.05 | 0.64 | 0.69 | 0.05 |
| Site7 | 0.70 | 0.75 | 0.05 | 0.73 | 0.70 | -0.03 |
| Site8 | 0.70 | 0.70 | 0 | 0.71 | 0.70 | -0.01 |
| Site9 | 0.64 | 0.78 | 0.14 | 0.65 | 0.78 | 0.13 |
| Site10 | 0.56 | 0.72 | 0.16 | 0.60 | 0.68 | 0.08 |
| Site11 | 0.82 | 0.84 | 0.02 | 0.81 | 0.82 | 0.01 |
| Site12 | 0.57 | 0.67 | 0.1 | 0.57 | 0.67 | 0.1 |
| Site13 | 0.59 | 0.73 | 0.14 | 0.64 | 0.73 | 0.09 |
| Site14 | 0.60 | 0.63 | 0.03 | 0.60 | 0.62 | 0.02 |
| Site15 | 0.65 | 0.68 | 0.03 | 0.65 | 0.67 | 0.02 |
| Site16 | 0.71 | 0.76 | 0.05 | 0.71 | 0.76 | 0.05 |
| Site17 | 0.64 | 0.64 | 0 | 0.64 | 0.65 | 0.01 |

shown in S6 Fig, the spatial distribution and overall pattern of case–control differences remained highly consistent, indicating that the biological signal was preserved. Moreover, the change in effect sizes was minimal, with the absolute difference |Δd| between pre- and post-ComBat estimates being less than 0.08 across edges.

(4) We conducted an ablation study over five candidate thresholds for graph construction (10%, 15%, 20%, 25%, and 30%) across all three evaluation schemes used in this study: (i) 10-fold cross-validation, (ii) leave-one-site-out cross-validation, and (iii) training on the principal dataset with testing on the independent validation cohort. The results are summarized in S7 Fig and Table E-F in S1 Text. Across all three evaluation schemes, the 20% threshold consistently achieved the highest or near-highest mean AUC, with only minimal fluctuations in performance at adjacent thresholds.

(5) we systematically evaluated the impact of GNN architecture on performance and explicitly report both cross validation and independent validation results for the requested variants. The full results are now summarized in S8 Fig and S9 Fig. Across both datasets and both evaluation schemes, the configuration with 5 layers and a 128-dimensional hidden space consistently achieved the best or near best performance, while the other settings showed only moderate fluctuations in accuracy and AUC.

(6) we added a learning curve analysis in which we vary the size of the training set and plot test accuracy as a function of training sample size (S10 Fig). The results show a monotonic increase in test accuracy as the number of training subjects grows, with no evidence of abrupt degradation at larger sample sizes.

(7) To test whether medication status affects the subtyping solution, we performed a sensitivity analysis in which medication was treated as a covariate. As shown in S11 Fig A–B, the analysis again yielded three subtypes, and subtype assignments were highly stable: fewer than 2% of participants changed subtype relative to the original solution. Additionally, we conducted an additional analysis restricted to medication-naïve participants and repeated the subtyping. As shown in S11 Fig C–D, this restricted sample likewise produced three subtypes, with subtype assignments differing from the full-sample solution for fewer than 2% of participants.

(8) We performed an ablation study to investigate the impact of the multi-term objective function on generalization performance. As shown in Table G in S1 Text, our findings reveal that omitting any individual loss component results in a significant degradation in performance, highlighting the critical contribution of each term to reconciling sparsity with predictive accuracy while enhancing generalization.

## 4. Discussion

The current study could largely promote the recent attempts of MDD redefining and subtyping according to neural mechanisms beyond observable symptoms by providing a novel ensemble hybrid deep-learning framework. Deep-learning approach serves as an extension of traditional machine-learning approach [54] and can automatically learn a latent high-level information from raw input data, making this method ideally suited to studying complex, subtle and scattered brain patterns [55,56]. However, the potential advantages of deep-learning methods have not been fully exploited in neuroimaging domains. Functional connectivity profiles represent nuanced structures of functionally-connected brain networks that were overlooked by previous studies that simply flattened functional connectivity profiles into a vector. To tackle this problem, we adopted GNN which represented functional connectivity profiles as graphs, thus could reserve the inherent structure of functional networks. In addition, although some previous attempts have been made to employ deep learning studies in discriminating patients with mental disorders and healthy controls [57–60], these models are based on 'black-box' algorithms which prevent the identification of underlying diagnostic decisions [61,62]. Our model was self-explaining classifier which furnished identification of the structure of subgraphs that were crucial for diagnostic decisions. In order to

obtain reliable and reproducible subtypes of MDD, we developed an unsupervised data-driven method which could automatically identify number of MDD subtypes.

Our classifiers achieved an average generalization accuracy of 73% for leave-one-site-out cross validation with REST-meta-MDD dataset collected from 17 independent sites. Moreover, our models achieved the highest generalization accuracy in SRPBS dataset using the REST-meta-MDD as the training data compared other baselines. Recently, there has been increasing attention on improving the generalization of the classifier. It is an emerging consensus that the generalization of a machine learning framework should be first proved before it can be considered as practical in clinical applications to ensure its reproducibility [16,18]. The validation of the generalization for a classifier was even considered as "a bare minimal requirement" for its clinical application [18]. More importantly, the generalization should be tested with independent datasets from multiple sites [16]. Frameworks that showed high generalization on the diagnosis of ASD and Alzheimer's disease have been established [61,63,64]. As for MDD, pioneering work by Yamashita and colleagues has attempted to develop an MDD classifier using logistic regression. They achieved an average accuracy of 66% with an external dataset [16]. In the current study, we used a leave-one-site-out cross validation in REST-meta-MDD and used Chinese-population-based data to predict Japanese-population-based data, which comprehensively tested the generalization of our algorithm. Across evaluation settings, our model showed the best generalization among baselines, supporting reproducibility across sites and countries. However, with an overall generalization accuracy of 73%, the results are encouraging but not yet sufficient for clinical deployment. The classifier should therefore be viewed as a research tool, pending further prospective validation, calibration and clinical utility testing.

Another major contribution of the current study is that we consistently identified three depression subtypes across two datasets. It is well known that MDD is not a unitary disease but a heterogeneous disorder since the patients present varied symptoms and respond divergently to treatment [5], suggesting that different biological mechanisms underlie different subtypes. The importance of identifying diagnostic biomarkers for MDD subtypes has long been recognized, but effective solutions remain unclear. Few attempts have been made while mixed results were reported. Drysdale and co-authors proposed a framework to identify MDD subtypes according to resting-state functional connectivity profiles based on canonical correlation analysis (CCA) and hierarchical clustering approaches. They found four neurophysiological subtypes using canonical correlation analysis (CCA) and hierarchical clustering approaches [5]. However, another recent study only identified 2 depression subtypes with CCA and K-means clustering algorithm [65]. It remains unaddressed whether consistent and reproducible MDD subtypes could be derived from different datasets in order to provide clinically applicable biomarkers. To test this hypothesis, we adopt the data-driven CFPN algorithm which automatically obtain the number of MDD subtypes. Intriguingly, we consistently found three MDD subtypes across two datasets. In addition, the functional connectivity signatures of the three subtypes were also shown to be highly reproducible. Patients of subtype1 mainly demonstrated impairments in a brain system encompassing DMN network. Patients of subtype 2 were associated with deficits in the frontostriatal network, while the third subtype were mainly associated with the Insula-Cingulum-Temporal network. Our findings were in line with previous study by Drysdale [5], where subtype 3 and 4 showed abnormal connectivity within frontostriatal network and subtype1 and 2 were associated with impairments in cingulate areas. In addition, similar deficits in DMN and subcortical network were observed in subtype2 from previous study by Wang et al. [65]. These results suggested that patients with MDD as a group share some traits and common features of subtypes, which have potential to obtain biomarkers used for clinical guidance. However, these subtypes demonstrated overlapping without clear distinction unlike our depression subtypes in two previous studies. The current study thus extended previous studies and showed that highly consistent and reproducible MDD subtypes could be obtained with two large multisite datasets. Our findings could promote the development of personalized diagnostic and treatment strategy.

The subtype-specific connectivity motifs identified by our framework may be viewed as candidate circuit level biomarkers with potential relevance to precision treatment in major depressive disorder. Because the edge level explanations are derived directly from the predictive model, these signatures provide an interpretable mapping between latent embeddings and

biologically meaningful neural circuits, rather than remaining as opaque model internal features. Building on this, the three reproducible subtypes suggest a hypothesis generating framework for individualized neuromodulation target selection. For the subtype characterized by default mode network and cerebellar coupling, cerebellar stimulation may represent a plausible candidate target. For the subtype marked by prominent insula, cingulate, and temporal involvement, targeting the insula or nodes within the salience network may be preferentially considered. For the frontostriatal subtype, the circuit profile is more consistent with conventional dorsolateral prefrontal cortex targeting, suggesting alignment with standard DLPFC based stimulation protocols. Importantly, these translational implications are hypothesis generating and require prospective interventional trials to determine whether subtype guided targeting improves treatment response. In parallel, EH-BrainGNN should be regarded as a decision support tool rather than a replacement for DSM-5 based assessment. Clinical judgment and symptom ratings would remain primary, whereas the model could provide an objective neuroimaging derived risk estimate together with interpretable circuit evidence to support clinical review. Before any real-world deployment, rigorous prospective multisite validation, calibration, and appropriate governance and regulatory oversight will be essential.

One limitation of the current study is that we could not evaluate whether different neurobiological subtypes show differential treatment response, as both REST-meta-MDD and SRPBS provide only baseline clinical and neuroimaging measures. Future prospective studies with longitudinal follow-up are needed to test whether subtype assignment predicts response to rTMS or pharmacotherapy, and to determine whether subtype informed targeting and stimulation parameters improve outcomes beyond one size fits all protocols. Several methodological uncertainties should also be considered. Our results may be sensitive to preprocessing choices, including nuisance regression (for example head motion), parcellation, and graph construction thresholds, which can influence functional connectivity estimates and downstream classification and subtyping. Moreover, scanner and protocol heterogeneity across sites may introduce residual confounding that is mitigated but not eliminated by leave one site out validation and cross cohort transfer. Finally, alternative model designs and hyperparameter settings could yield different performance and subtype structure, motivating broader benchmarking and prospective validation in independent cohorts. In addition, unmeasured factors not captured in available metadata, including cultural or environmental influences, may contribute to cross population differences; thus, claims of biological equivalence across cohorts should be interpreted cautiously.

## Supporting information

**S1 Text. Supplementary Tables.**
(DOCX)

**S1 Fig. Sample Selection of REST-meta-MDD consortium.** From 2428 subjects, 1604 subjects were selected through above criteria.
(TIF)

**S2 Fig. Generation procedure of graph embeddings and t-SNE plot of FC network and graph embeddings.** (A) FC networks from REST-meta-MDD cohort of 1604 participants were used as inputs and a two-dimensional plot was generated using the t-SNE, where the dark green represented the patients with MDD and the gray represented healthy controls. (B) Procedure of obtaining the graph embeddings from node embeddings in a single participant is depicted. Total_latent_dim = the number of nodes + the number of hidden units × (the number of GNN layers -1). (C) Permutation distribution of the estimate using the trained EH-BrainGNN model (repetition times: 1000) are shown, where x- and y-labels represent the generalization rate and probability density. $GR_0$ denotes the generation rate gained by the EH-BrainGNN model trained on the real class labels. (D) Graph embeddings that served as inputs were embedded in a two-dimensional plot generated using the t-SNE for the two classes (MDD and HC). (E) The kernel distributions of graph embeddings of the MDD and HC populations are depicted.
(TIF)

**S3 Fig. Schematic diagram of the procedure for leave-one-site-out analysis.**
(TIF)

**S4 Fig. The stability of the subtype-specific top-20 edges.** (A) the stability of the subtype-specific top-20 edges across bootstrap resamples quantified by the Jaccard index. (B) the bootstrap selection frequency of the reference top-20 edges for each subtype.
(TIF)

**S5 Fig. The scatter plots of predicted vs observed HAMD scores.**
(TIF)

**S6 Fig. Cohen's *d* heatmaps for the raw FC and the ComBat-adjusted FC.**
(TIF)

**S7 Fig. The impact of graph construction threshold on 10-fold cross-validation.**
(TIF)

**S8 Fig. The impact of GNN layers and hidden dimensions on 10-fold cross-validation performance.**
(TIF)

**S9 Fig. The impact of GNN layers and hidden dimensions on generalization performance in the independent validation dataset using the principal dataset.**
(TIF)

**S10 Fig. Learning curve analysis.**
(TIF)

**S11 Fig. The impact of medication status on subtype analysis.** (A-B) CFDP decision graph and spatial distribution of subtypes with medication included as a covariate. (C-D) CFDP decision graph and spatial distribution of subtypes with medication-naïve participants.
(TIF)

## Author contributions

**Conceptualization:** Kaizhong Zheng, Badong Chen, Baojuan LI.

**Data curation:** Kaizhong Zheng, Huaning Wang.

**Funding acquisition:** Badong Chen, Baojuan LI.

**Methodology:** Kaizhong Zheng, Dewen Hu, Badong Chen, Baojuan LI.

**Software:** Kaizhong Zheng.

**Supervision:** Kaizhong Zheng, Baojuan LI.

**Validation:** Hongbing Lu, Badong Chen.

**Writing – original draft:** Kaizhong Zheng, Baojuan LI.

**Writing – review & editing:** Kaizhong Zheng, Hongbing Lu, Huaning Wang, Dewen Hu, Badong Chen, Baojuan LI.

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
