## [Decision Letter · Decision Letter 0]

28 Sep 2025

Response to Reviewers
Revised Manuscript with Track Changes
Manuscript
**Journal Requirements:**

1. Please provide a detailed online Financial Disclosure statement. This is published with the article. It must therefore be completed in full sentences and contain the exact wording you wish to be published.

a) State the initials, alongside each funding source, of each author to receive each grant, if applicable. For example: “This work was supported by the National Institutes of Health (####### to AM; ###### to CJ) and the National Science Foundation (###### to AM).”

For more information, please see our guidelines: https://journals.plos.org/digitalhealth/s/submission-guidelines#loc-financial-disclosure-statement

2. Please ensure that the funders and grant numbers match between the Financial Disclosure field and the Funding Information tab in your submission form. Note that the funders must be provided in the same order in both places as well.

3. Please update your online Competing Interests statement. If you have no competing interests to declare, please state: “The authors have declared that no competing interests exist.”

4. Please ensure that the Title in your manuscript and the Title in your online submission form are the same.

5. Please provide separate main figure files in .tif or .eps format only and ensure that all files are under our size limit of 10MB. You may leave the embedded figures in the manuscript.

For more information about how to convert your figure files please see our guidelines: https://journals.plos.org/digitalhealth/s/figures

6. We have noticed that you have uploaded Supporting Information files, but you have not included a list of legends. Please add a full list of legends for your Supporting Information files before or after the references list.

7. Some material included in your submission may be copyrighted. According to PLOS’s copyright policy, authors who use figures or other material (e.g., graphics, clipart, maps) from another author or copyright holder must demonstrate or obtain permission to publish this material under the Creative Commons Attribution 4.0 International (CC BY 4.0) License used by PLOS journals. Please closely review the details of PLOS’s copyright requirements here: PLOS Licenses and Copyright. If you need to request permissions from a copyright holder, you may use PLOS's Copyright Content Permission form.

Potential Copyright Issues:

Figure 2: Please confirm whether you drew the images / clip-art within the figure panels by hand. If you did not draw the images, please provide (a) a link to the source of the images or icons and their license / terms of use; or (b) written permission from the copyright holder to publish the images or icons under our CC-BY 4.0 license. Alternatively, you may replace the images with open source alternatives. See these open source resources you may use to replace images / clip-art:

- https://openclipart.org/

**Additional Editor Comments (if provided):**

Thank you for your submission. The study is promising and of broad interest, but as echoed across all reviews key aspects of methodological transparency, leakage control, interpretability, and robustness require clarification or additional analyses before the work can be considered further.

Please provide a point-by-point response to all reviewer comments and these editor items, and revise the manuscript accordingly.

1. ComBat usage and leakage control

* Explicitly state, for every evaluation setting (10-fold CV, leave-one-site-out, and independent validation), that ComBat parameters were estimated only on the training data within each outer fold/site and then applied to the held-out fold/site without re-fitting. For the cross-national test, fit on the Chinese training sites only and apply those parameters to the Japanese sites; confirm no parameters were estimated using any Japanese labels or distributions. Add a brief schematic of the preprocessing/CV pipeline that shows where ComBat is fit/applied. See Methods “Control of site differences and covariates” and Results sections describing CV/LOSO (e.g., pages 10–11 and 16–18), and make these procedures explicit.

2. Hyperparameter search protocol

* In “Training, testing and hyperparameter optimization,” specify the exact inner-loop search (e.g., inner K-fold within each outer fold), the grid explored (Table 2), and the selection criterion (e.g., mean AUC across inner folds). Confirm that hyperparameters and early-stopping criteria were fixed before scoring any outer test fold or the independent cohort. Provide code-level details needed to reproduce the search. Tie Table 2 to this description and report the final selected values per experiment.

3. Sensitivity analyses for deep model capacity and data size

* Report performance when varying the number of GNN layers and hidden dimensions (at minimum the values already listed in Table 2). Provide both cross-validation and independent-validation results for these variants. Add a learning-curve analysis showing accuracy/AUC as a function of training sample size to support the claim of robust representation learning.

4. Graph construction threshold (top 20%)

* Justify the 20% binarization choice with citations; if empirical, report an ablation across thresholds (e.g., 10/15/20/25/30%) and summarize the impact on performance and subtype discovery. Consider treating the threshold as a tunable parameter in inner CV and report whether conclusions are stable.

5. From embeddings to edges: interpretability path

* Figures 5–6 show subtype-associated edges but the derivation from learned embeddings is not sufficiently documented. Expand Methods to detail how edges are selected from the explanation mask/embeddings, including any statistics, thresholds, or aggregation across subjects. Discuss interpretability in the context of biomarker development.

6. Stability and significance of subtype-specific patterns

* Quantify the stability of CFDP subtypes and the “top 20 edges” using resampling/bootstrapping (e.g., adjusted Rand index/NMI across resamples, edge selection frequencies, and permutation tests with multiple-comparison control). Report effect sizes and confidence intervals where appropriate.

7. CFDP decision rules and K selection

* In “Module D: Subtype generator,” report the explicit rules or thresholds used on the CFDP decision graph to choose cluster centers and the number of clusters. Provide the decision graph(s) and show why K=3 is selected in both datasets, with a cluster-quality metric (e.g., silhouette, Dunn, or density-based criteria).

8. Clinical covariates and medication

* REST-meta-MDD contains clinical variables (e.g., medication). Clarify availability and whether medication status (and other clinical covariates) affect subtyping. Provide sensitivity analyses controlling for medication (and, if feasible, analyses restricted to medication-naïve participants), or clearly justify if unavailable.

9. Symptom prediction statistics

* In “Correspondences between functional signature and MDD symptoms,” report significance testing for r² values (e.g., permutation-based P values with FDR control), 95% CIs, and add the requested scatter plots of predicted vs observed scores. Clarify that “leave-one-out ridge regression” is leave-one-subject-out CV (not leave-one-site-out). Report the exact number of PCA components retained (not only “95% variance”).

10. Statistical reporting and comparators

* Where you mark superiority (e.g., Fig. 3A–B), specify the statistical test, correction for multiple comparisons, and report exact P values or adjusted CIs. Provide 95% CIs for key metrics (ACC/AUC), site-wise confusion matrices or balanced accuracy where class imbalance exists, and a brief calibration summary (e.g., Brier score or reliability plot) for the classifier.

11. Reproducibility resources

* Provide a public repository with full code, random seeds, environment details, and scripts to reproduce preprocessing, ComBat fitting/applying within CV, model training, and all analyses/figures. Include model weights or instructions to retrain.

12. Overall framework exposition

* Rewrite “Overall framework of EH-BrainGNN” to first present representation learning for classification, then the downstream subtyping on learned features. Add a schematic of the full network with parameter counts and a clear statement of the role of the subgraph/explainer (see Fig. 2 on page 25 and related text).

13. Figures and captions

* Figure 3C vs text: ensure consistency between “ranking” and “accuracy,” or present both in a harmonized way.

* Figure 5D: thicken lines/edges and improve visibility (Reviewer 2).

* Add the requested scatter plots for symptom prediction.

* Correct any caption mismatches (e.g., the caption noted by Reviewer 2 around lines 505–507) and typographical slips in Results (e.g., “subtyp2s”). Indicate per-subtype sample sizes alongside percentages.

**Reviewers' Comments:**

Reviewer #1: 1.Please clarify the scope of ComBat's application. For instance, in lines 280-281, the sentence“Furthermore, we used ComBat harmonization method to mitigate distinct site differences and artifact or confounding effect”should specify whether ComBat was fitted on each outer training fold and then applied to that fold's test set.

2.The section“Training, testing and hyperparameter optimization”did not specify the grid search process or the basis for hyper-parameter selection. For instance, was the grid search conducted on the inner layers of training set in each outer fold? Was the selection criterion based on maximizing the average AUC of the inner layers?

3.For lines 149-150”To create A, we binarized the FC matrix by transforming only the top 20-percentile absolute correlation values into ones, and the rest were transformed into zeros.”, please explain the rationale for selecting the 20% threshold (citing relevant literature). If it was empirically determined, please provide justification. Alternatively, it is suggested to test other thresholds and report their impact on results.

4.Please describe the stability of CFDP results, or provide evidence of the stability and statistical significance of the subtype-specific explanatory edges (i.e., the”top 20 edges”mentioned in line 252).

5.In the section“Module D: Subtype generator,”please supplement with an explicit threshold or rule used for selecting centers and determining the number of clusters on the decision graph. If possible, please report the evidence for obtaining three subtypes under this rule.

6.Considering that the REST-meta-MDD dataset includes clinical information such as medication status, please clarify whether these factors influence the subtyping results.

7.In the section”Correspondences between functional signature and MDD symptoms,”the authors only reported R2. It is recommended to also provide corresponding significance levels.

8.Please unify the terminology of abbreviations (e.g., SBN in Figure 5 and SCN in Table S3).

Reviewer #2: The authors proposed a novel ensemble hybrid deep-learning framework for brain network using the graph neural network (which they call EN-BrainGNN). They applied their framework to MDD classification and subtyping task in the multi-site and cross-nation setting using two large MDD database (Rest-META-MDD, N=1604, China and SRPBS, N=446, Japan). In the supervised classification task, it achieved 72% accuracy in the independent validation dataset showing the strong across-site and cross-national generalization. In the unsupervised clustering task, they identified three reproducible biological subtypes across two datasets.

This study is an important contribution demonstrating that representation learning with deep learning models for MDD biomarker development can generalize across multi-site and cross-national datasets, and is applicable to both classification and subtyping. In previous biomarker studies using functional connectivity (FC) as features, progress had been made in achieving multi-site generalization for classification and subtyping. However, the performance had nearly reached an upper bound, and changing classification algorithms alone did not yield significant improvements. On the other hand, many studies employing deep learning lacked evaluation with independent datasets. This study showed, using the independent validation dataset, that graph embedding features learned by deep learning not only improve classification accuracy but also enable reproducible subtyping. Thus, it represents an important step toward applying deep learning frameworks to clinical research.

Although the results presented in this paper are highly significant, several concerns remain. First, the explanation of the deep learning framework is insufficient. Second, there is no description of methods for interpreting the learned features. Third, an analysis is needed to clarify why deep learning, which typically involves a large number of parameters and is prone to overfitting, was able to achieve such robust representation learning. Addressing these questions would help bridge the gap between the proposed interpretable deep learning–based representation learning method and conventional machine learning approaches that use FC features.

[Major comments]

1. The section Overall framework of EH-BrainGNN is extremely difficult to follow. To improve readability, a thorough rewrite is necessary. For example, it would be clearer to first describe feature representation learning through the classification task, and then explain the downstream subtyping task using the learned features. A schematic diagram of the overall deep network architecture, including the number of model parameters, is also needed. In addition, the role of the subgraph is not explained sufficiently and should be clarified.

2. In Figures 5 and 6, the authors present edges associated with the three subtypes. However, it is not at all clear how these edges were derived from the learned embedding features. Please provide detailed explanation in the Methods section, and discuss the importance of interpretability in the deep learning framework for biomarker development.

3. Although the best hyperparameters are reported, the sensitivity of the results to different parameter choices is not examined. In particular, the number of GNN layers and the hidden dimensions are likely to be the two most important parameters. Please provide classification performance results (both cross-validation and independent validation) by varying these two parameters. From another perspective, it is also essential to consider how many samples are required to obtain robust features using deep learning. Please compare classification performance while varying the number of training samples.

[Minor comments]

1. [line 150] Please add explanation of the choice of top 20%-tile.

2. [line 151] By definition, rho(k,k) is always one for any k. Does it cause any problem ?

3. [line 255] What does “leave-one-out ridge regression model” mean ? Does it mean leave-one-site-out cross validation using the ridge regression model ?

4. [line 257] How many dimensions after PCA ?

5. [line 277] How did the authors determine the final setting? Were these parameters fixed without looking at the independent validation data? If not, overfitting can be expected to some extent. Please clarify and discuss.

6. [line 279] Did the authors apply the identical procedure (network architecture, hyperparameters, harmonization) to two different datasets independently? Please clarify.

7. [line 287] The description about Fig.3(a) seems to be incorrect.

8. [line 292-293] Figure 4 (B) showed clear separation of MDD and HCs. If this is true, the classification performance should be much higher. Why is this not the case ?

9. [line 298-299] Did the authors also apply resulting parameters to test data to harmonize test data?

10. [line 300-301] In Figure 3(c), the plot shows ranking, whereas the text describes accuracy. This discrepancy is confusing. Please unify the presentation—either by reporting both as accuracy or both as ranking.

11. [line 310-311] The same as the comment 9 above. Did the authors also apply resulting parameters to the independent validation data to harmonize them?

12. [line 323] The details of CFDP algorithm is lacking. This method is known to be sensitive to the cut-off parameters to estimate density and distance. How to automatically determine the number clusters is not explained. Please add explanation in the method section.

13. [line 337-338] Lines in Figure5D are too thin and weak to see. Please increase visibility of the graphs.

14. [line 364] There are not scatter plots in the manuscript. Actually scatter plots are very helpful to understand how accurate it is. Please add the scatter plots in the manuscript.

15. [line505-507] The caption seems to be wrong.

Reviewer #3: This is a timely and ambitious study that applies graph neural networks and clustering to classify and subtype MDD across large multi-site datasets. The technical design is strong, and the effort to test generalization across sites and even across countries is a real strength. That said, the work is not yet ready for publication without further validation. Reported accuracy is encouraging but not sufficient for clinical use, the subtyping results need stronger benchmarking and replication, and some key modeling choices require sensitivity checks. The discussion also needs to frame the findings more cautiously and acknowledge limitations more fully.

I see this as a promising contribution but recommend major revision. I provide detailed feedback in my review to guide the authors on what to improve.

Reviewer #4: This manuscript addresses an important and timely question in precision psychiatry: whether resting-state functional connectivity, analyzed through an ensemble graph neural network (EH-BrainGNN) with clustering, can improve classification and subtyping of major depressive disorder (MDD). The work has notable strengths, including use of large multi-site datasets (REST-meta-MDD, N=1604; SRPBS, N=446), validation across Chinese and Japanese cohorts, and the attempt to identify reproducible biological subtypes of MDD. However, the manuscript has significant weaknesses in presentation, methodological transparency, statistical analysis, and interpretation. These must be addressed before the paper could be considered for publication in a plos dogital health journal.

**Figure resubmission:**

**Reproducibility:** To enhance the reproducibility of your results, we recommend that authors of applicable studies deposit laboratory protocols in protocols.io, where a protocol can be assigned its own identifier (DOI) such that it can be cited independently in the future. Additionally, PLOS ONE offers an option to publish peer-reviewed clinical study protocols. Read more information on sharing protocols at https://plos.org/protocols?utm_medium=editorial-email&utm_source=authorletters&utm_campaign=protocols

---

## [Decision Letter · Decision Letter 1]

8 Feb 2026

Functional connectivity–based classification and subtyping of major depression for precision mental health: An ensemble graph neural network approach

PDIG-D-25-00628R1

Dear Dr. LI,

We are pleased to inform you that your manuscript 'Functional connectivity–based classification and subtyping of major depression for precision mental health: An ensemble graph neural network approach' has been provisionally accepted for publication in PLOS Digital Health.

Best regards,

Phat Kim Huynh, Ph.D.

Guest Editor

PLOS Digital Health

**Additional Editor Comments (if provided):**

**Reviewer Comments (if any, and for reference):**

Reviewer's Responses to Questions

**Comments to the Author**

Reviewer #1: All comments have been addressed

Reviewer #3: All comments have been addressed

Reviewer #4: All comments have been addressed

publication criteria?

Reviewer #1: Yes

Reviewer #3: Yes

Reviewer #4: Yes

3. Has the statistical analysis been performed appropriately and rigorously?

Reviewer #1: Yes

Reviewer #3: Yes

Reviewer #4: Yes

4. Have the authors made all data underlying the findings in their manuscript fully available (please refer to the Data Availability Statement at the start of the manuscript PDF file)?

Reviewer #1: Yes

Reviewer #3: Yes

Reviewer #4: Yes

5. Is the manuscript presented in an intelligible fashion and written in standard English?

Reviewer #1: Yes

Reviewer #3: Yes

Reviewer #4: Yes

Reviewer #1: (No Response)

Reviewer #3: The authors have addressed my earlier concerns about methodological clarity and stability. I have no additional comments beyond what I mentioned in my previous review.

Reviewer #4: The revised manuscript reflects a careful and substantial response to the reviewers’ comments, with clear improvements in methodological transparency, robustness analyses, and overall rigor. A few issues remain before acceptance: the abstract should better benchmark the reported accuracy against prior studies and more clearly quantify generalization across sites and countries; interpretability and subtyping results, while biologically plausible, remain largely qualitative and should be framed more explicitly as exploratory or supported with simple quantitative comparisons; and statements regarding clinical or neuromodulatory implications should be further tempered to avoid overinterpretation. In addition, the authors are encouraged to emphasize the method- and cohort-dependence of subtype solutions and clarify reproducibility details such as model weights and random seed control. Addressing these points should render the manuscript suitable for publication.

**Do you want your identity to be public for this peer review?** For information about this choice, including consent withdrawal, please see our Privacy Policy

Reviewer #1: **Yes:** Shao-Wei Xue

Reviewer #3: **Yes:** Victor Ifechukwude Agboli

Reviewer #4: **Yes:** Akbar Ali
